# Speculations on the application of foliar $^{13}$C discrimination to reveal groundwater dependency of vegetation, provide estimates of root depth and rates of groundwater use

Rizwana Rumman[1], James Cleverly[1], Rachael H. Nolan[1], Tonantzin Tarin[1] and Derek Eamus[1,2]

[1]Terrestrial Ecohydrology Research Group

School of Life Sciences

University of Technology Sydney

PO Box 123

Broadway

NSW 2007

Australia

*Correspondence to*: Derek Eamus  (Derek.Eamus@uts.edu.au)

**Abstract.**

Groundwater-dependent vegetation is globally distributed, having important ecological, social and economic value. Along with the groundwater resources upon which it depends, this vegetation is under increasing threat through excessive rates of groundwater extraction.

In this study we examined one shallow-rooted and two deep-rooted tree species at multiple sites along a naturally occurring gradient in depth-to-groundwater.  We measured (i) stable isotope ratios of leaves ($\delta^{13}$C), xylem and groundwater ($\delta^2$H and $\delta^{18}$O); and (ii) leaf vein density. We established that foliar discrimination of $^{13}$C ($\Delta^{13}$C) is a reliable indicator of groundwater use by vegetation and can also be used to estimate rooting depth. Through comparison with a continental-scale assessment of foliar $\Delta^{13}$C, we also estimated the upper limits to annual rates of groundwater use. We conclude that maximum rooting depth for both deep-rooted species ranged between 9.4 m and 11.2 m and that annual rates of groundwater use ranged *ca* 1400 – 1700 mm for *Eucalyptus camaldulensis* and 600 – 900 mm for *Corymbia opaca*. Several predictions about hydraulic and leaf traits arising from the conclusion that these two species made extensive use of groundwater were supported by additional studies of these species in central Australia.

## 1 Introduction

Drylands cover 41% of the earth's total land area (Reynolds et al., 2007) and are sub-categorized as hyper-arid, arid, semi-arid and dry sub-humid areas. Hyper-arid, arid and semi-arid regions are characterized by chronic water shortage with unpredictable rainfall (Clarke, 1991). Approximately 40 % of the world's population reside in drylands and groundwater represents a major water resource both for human consumptive use but also for groundwater dependent ecosystems (GDEs' Eamus et al., 2005). Sustainable management of both groundwater and GDEs requires identification of the location of GDEs, rooting depth of vegetation and rates of groundwater use but attaining such information present significant technical and cost challenges (Eamus et al., 2015).

Approximately 70% of Australia is classified as semi-arid or arid (Eamus et al., 2006; O'Grady et al., 2011). Furthermore, annual potential evaporation exceeds annual rainfall across most of the continent, thus most Australian biomes are water-limited according to the Budyko (1974) framework (Donohue et al., 2009). On average, central Australia receives less than 350 mm y$^{-1}$ of rainfall, making water a primary limiting resource (Eamus et al., 2006). Because surface water bodies in this region are mostly ephemeral (NRETAS, 2009, although see Box et al., 2008, regarding the small number of permanent water bodies), groundwater plays an important role in maintaining ecosystem structure and function of terrestrial (especially riparian) vegetation (Eamus et al., 2006). Owing to the remoteness of much of Australia's interior, few studies have investigated groundwater use by vegetation communities in these semi-arid regions.

Stomatal conductance is regulated to maximise carbon gain whilst simultaneously minimising transpiration (Cowan and Farquhar, 1977; Medlyn et al., 2011) and is sensitive to both soil and atmospheric water content (Prior et al., 1997; Thomas and Eamus, 1999). Intrinsic water-use efficiency (WUE$_i$), defined by the ratio of carbon gain to stomatal conductance) provides valuable insights on how vegetation responds to variation in water availability (Beer et al., 2009). Declining water supply results in increased WUE$_i$ as stomatal conductance declines (Eamus et al., 2013). Discrimination against the $^{13}$C isotope ($\Delta^{13}$C) is commonly used to calculate WUE$_i$. $\Delta^{13}$C provides a time-integrated measure of WUE$_i$ (Cernusak et al., 2011); in this study we examined spatial and seasonal patterns in $\Delta^{13}$C across three tree species in the Ti Tree basin.

The present study was undertaken in the Ti Tree basin, which is the location of an important groundwater resource in central Australia (Cook et al., 2008a). Rainfall occurs mostly in large events during the austral summer (Dec – Mar); thus there is minimal rainfall available for vegetation use over prolonged periods. The dry-season in this region is characterized by declining soil water availability and high vapour pressure deficits (Eamus et al., 2013). Previous studies have documented several surprising attributes for a number of tree species in the Ti Tree. O'Grady et al.,. (2009) observed that, despite living in an extremely water limited environment, the specific leaf-area (SLA) of *Corymbia opaca* and *Eucalytptus camaldulensis* were similar more to those from highly mesic environments than to species from arid environments. Similarly, Santini et al.,. (2016) observed that xylem wall thickness and vessel implosion resistance were significantly smaller in *E. camaldulensis*

and *C. opaca* than in shallow-rooted *Acacia aneura*. Finally, rates of water use and changes in midday water potential between the end of the wet-season and the end of the dry-season were minimal for *E. camaldulensis* and *C. opaca* but were very large for *Acacia aptaneura* (which was previously classified as *Acacia aneura*; Maslin and Reid, 2012; Nolan et al., 2017). *A. aneura* and *A. aptaneura* often intermix with other members of the large Mulga complex of closely related *Acacia*
species (Wright et al., 2016), thus we will refer *Acacia* spp. in the Mulga complex by the primary type, *A. aneura*. *A. aneura* is shallow-rooted and associated with shallow hard pans in this catchment (*ca* 1 m below ground surface; Cleverly et al., 2016a; Cleverly et al., 2016b) which prevents access to the groundwater below. *E. camaldulensis* is a riparian species, confined to narrow corridors along the ephemeral streams in the Ti Tree (the Woodforde River and Allungra Creek) where groundwater depth is shallow (< 3m); *C. opaca* is deep- rooted and may access groundwater to depths of 8 m or more
(O'Grady et al., 2009). These observations lead to the first hypothesis tested in the present study: that $WUE_i$ of *A. aneura* would be significantly larger than that of *E. camaldulensis* and *C. opaca* because reliance on shallow stores of water by Mulga imposes severe restrictions on water use, thus resulting in a large $WUE_i$.

Vertical and horizontal distance from rivers receiving groundwater inflows in arid-zones influences the degree to which trees access groundwater (O'Grady et al., 2006a; Thorburn et al., 1994). Trees closest to the river (vertically and horizontally)
have xylem water deuterium and $^{18}O$ isotope ratios ($\delta^2H$ and $\delta^{18}O$, respectively) that are close to the ratios of river and groundwater; trees further from the river have xylem water ratios increasingly different from those of groundwater and the river (O'Grady et al., 2006a). In endoreic basins like Ti Tree, evaporation of near-surface soil water imposes additional fractionation of $\delta^2H$ and $\delta^{18}O$ (Craig, 1961) relative to groundwater, thus $\delta^2H$ and $\delta^{18}O$ in xylem provides information on plant water source and climate (Cullen and Grierson, 2007). Variation in plant water sources with distance from the river can
affect stomatal conductance and $WUE_i$. In this study we tested the hypothesis that $WUE_i$ would increase with distance from the creek.

Differential access to water among co-occurring species within a biome results in variation of several morphological traits, including SLA (Warren et al., 2005), Huber value (Eamus et al., 2000; Sperry, 2000) and wood density (Bucci et al., 2004; Hacke et al., 2000). Leaf vein density (LVD) is a trait that influences whole-plant performance. From a resource investment
perspective, leaves are composed primarily of two components: mesophyll that undertakes photosynthesis and a leaf-vein network which delivers water and nutrients to the leaf. Investment in leaf veins is underpinned by resource allocation strategies (Niinemets et al., 2007; Niinemets et al., 2006; Niklas et al., 2007). Leaf-vein density is responsive to several environmental variables, but especially aridity (Uhl and Mosbrugger, 1999). Furthermore, LVD is positively correlated with leaf hydraulic conductance ($K_{leaf}$), maximum photosynthetic rate and leaf-level gas-exchange rates (Brodribb et al., 2007;
Sack et al., 2003; Sack and Frole, 2006; Sack and Holbrook, 2006). Consequently we investigate whether investment in LVD of three dominant overstorey tree species was affected by increasing depth-to-groundwater (DTGW).

The propensity for leaves to lose water matches the capacity of xylem to deliver the same volume of water (Brodribb and Holbrook, 2007; Meinzer and Grantz, 1990; Sperry, 2000), and positive correlations consequently occur between leaf-hydraulic conductance ($K_{leaf}$) and LVD (Brodribb et al., 2007; Sack and Holbrook, 2006). LVD provides a direct estimate of $K_{leaf}$ because it correlates with the distance water must traverse from termini of the xylem-network to sites of evaporation (Brodribb et al., 2010). Since transpiration is directly linked to availability of water to roots, we hypothesised that LVD will be correlated with depth-to-groundwater in plants for which groundwater is accessible; this correlation should be absent in species with shallow roots which cannot access groundwater. Whilst a number of studies have demonstrated increased LVD with increasing aridity along rainfall gradients (Brodribb et al., 2010; Brodribb and Holbrook, 2003; Sack and Holbrook, 2006), this relationship has not, to our knowledge, been examined in relation to DTGW. Finally, because LVD is strongly correlated with $K_{leaf}$ and rates of leaf-scale gas exchange (Brodribb et al., 2007; Sack et al., 2003; Sack and Frole, 2006), we hypothesise that LVD will be significantly correlated with $\Delta^{13}C$ (and hence $WUE_i$).

To summarize, we address the following questions:

1. Does access to groundwater by *E. camaldulensis* and *C. opaca* result in significantly smaller $WUE_i$ compared to *A. aneura* (Mulga)?
2. Does LVD correlate with DTGW in the three species examined?
3. Is there a correlation between LVD and $\Delta^{13}C$ (and hence $WUE_i$) for the three species examined?
4. Does horizontal and vertical distance from a known river flood-out zone influence foliar $\Delta^{13}C$ and $WUE_i$ of co-occurring species?
5. Can foliar $\Delta^{13}C$ be used as an indicator of utilisation of groundwater by vegetation of arid regions?
6. Can foliar $\Delta^{13}C$ be used to estimate rooting depth and upper and lower bounds of rates of groundwater use?

## 2 Materials and methods

### 2.1 Site description

The study was conducted in the Ti Tree Basin, a 5500 km$^2$ basin located approximately 200 km north of Alice Springs NT and 180 km north of the Tropic of Capricorn (22.28°S, 1933.25°E, 549 m asl). Climate is characterized as tropical and arid with hot summers and warm winters. The nearest Bureau of Meteorology station (Territory Grape Farm; Met Station 015643; within 25 km of all study sites) recorded mean and median annual precipitation of 319.9 and 299 mm, respectively (1987 - May 2016; http://www.bom.gov.au/). Of the annual median rainfall, 72% falls during the summer months (December - February) and 86% falls during the monsoon season (November - April). Mean minimum and maximum monthly temperatures range from 5°C and 22.6°C in July to 22°C and 37.5°C in January.

The soil is a "red kandosol" (74:11:15 sand: silt: clay; Eamus et al., 2013), typical of large portions of semi-arid Australia, and has a high potential for drainage (Morton et al., 2011; Schmidt et al., 2010). Patches of hard siliceous soil is often observed and are likely surface expressions of the underlying hardpan (Cleverly et al., 2013), a common formation in the top $1 - 1.5$ m in this type of soil (Cleverly et al., 2013; Cleverly et al., 2016a; Cleverly et al., 2016b; Morton et al., 2011). The

major potable source of water for this region is a large underground reservoir, recharged mainly by seepage from creek/river channels and their flood-out zones, "mountain" front recharge and by occasional very heavy rainfall events (NRETAS, 2009; Calf et al., 1991). The Ti Tree basin has a natural gradient in DTGW. The depth of the water table below ground level is shallow ($< 2$m) in the northern part and groundwater is lost through evapotranspiration (Shanafield et al., 2015), whereas DTGW reaches 60 m in the southern and western part of the basin and $20 - 40$ m in the eastern region (NRETA, 2007).

All study sites were characterized as being in one of three distinct vegetation types (Nolan et al., 2017; Cleverly et al., 2016a): (1) riparian, predominantly consisting of *Eucalyptus camaldulensis var. obtusa*, which line the banks of the ephemeral streams in the Ti Tree basin (Woodforde River and Allungra Creek); (2) low mixed Mulga woodland (*A. aneura* F.Muell. ex Benth., *A. aptaneura* Maslin & J.E.Reid, *A. kempeana* F.Muell.) with an understorey of shrubs, herbs and $C_3$ and $C_4$ grasses; (3) tall, open *Corymbia* savanna with extensive Spinifex grass (*Triodia spp.*), sparse *Corymbia opaca* (D.J.Carr

& S.G.M.Carr) K.D.Hill & L.A.S.Johnson trees and occasional Mulga trees. The Woodforde River and Allungra Creek are ephemeral streams that flow only after large extensive rainfall events. Nonetheless, perched aquifers beneath their riparian corridors are recharged by large storms (Villeneuve et al., 2015), providing long-term access to groundwater near ephemeral streams. Allungra Creek and its flood-out zone represent zones of local recharge (NRETAS 2009). Overbank flooding and sheet flow occur in flood-outs where the Woodforde River and Allungra Creek enter the basin and split into a network of

smaller channels (NRETA, 2007), resulting in an estimated 1.8 ML of groundwater recharge per year (NRETAS, 2009).

Four sampling plots were established for determination of foliar $\Delta^{13}C$ and of $\delta^{18}O$ and $\delta^{2}H$ values for xylem water and groundwater from a nearby bore. DTGW in each of the four plots was 4.4 m, 8.3 m, 8.8 m and 13.9 m, respectively. One of the four plots was located on the banks of the Woodforde River (plot 1, DTGW = 4.4 m). *E. camaldulensis* is the dominant tree species in plot 1, and *C. opaca* is also present. In the second plot, *A. aneura* is the dominant species (plot 2, DTGW =

8.3 m). *C. opaca* is the dominant tree species in the two remaining plots (plot 3, DTGW = 8.8 m; plot 4, 13.9 m).

$\Delta^{13}C$ of *E. camaldulensis*, *C. opaca* and *A. aneura* were examined along three additional transects to investigate the influence topography and of distance from a creek on $WUE_i$. The three transects established perpendicular to the banks of Allungra Creek, which is in an area of known groundwater recharge at the base of the hills that bound the southern extent of the basin (NRETAS, 2009). One of the three transects was located near a permanent waterhole near the flood-out and the

bottom of Allungra Creek. Transects two and three were $1 - 2$ km upstream of the water hole. Transects extended from the shore within 1 m of the creek up to a maximum of 1800 m from the creek (vertical distance from the creek bed ranged $0 - 4$

m; Supplementary Fig S2a –c), across which vegetation graded from riparian forest to *Corymbia* open savanna, with occasional Mulga trees interspersed throughout.

In addition to the four plots and three transects, "spot sampling" for foliar $\Delta^{13}C$ alone was performed at three sites to extend the examination of variation in WUE$_i$ to   20 m, 36 m and 49.5 m DTGW for *A. aneura* or 20 m and 36 m for *C. opaca*. Finally, continental sampling of foliar $\Delta^{13}C$ of multiple dominant tree species was undertaken at seven sites distributed across Australia (Rumman et al., 2017) (Table S1; Karan et al., 2016; Rumman 2017) in order to allow comparison of foliar $\Delta^{13}C$ of our three Ti-Tree species with a continental-scale regression of $\Delta^{13}C$ with rainfall.

**2.2 Leaf sampling protocols and meteorology**

Sampling was undertaken in April 2014 (end of wet-season) and September 2013 (end of dry-season). Three mature, healthy leaves on each of three branches from two or three replicate trees were sampled for $\Delta^{13}C$ in all plots, transects and spot sampling sites.  In addition, terminal branches of the trees in the four plots were collected for deuterium and $\delta^{18}O$ analyses of xylem water. Bore water samples were also collected in the four plots and the three spot sampling sites using a Groundwater samples were also collected from the bores located at each site using HydraSleeve no-purge groundwater sampler (Cordry, 2003). Sampling along the three transects occurred at three or four points along each transect. The three trees of each of the dominant species at each location were located within 50 m of the bore. Leaves for leaf vein analysis (see below) were collected during Sept 2013.

Climate conditions preceding and during the sampling periods were obtained from two eddy-covariance towers (Fluxnet sites AU-ASM and AU-TTE; Cleverly et al., 2016a; Cleverly 2011; Cleverly 2013). AU-ASM is located to the west of the Ti Tree at the spot sampling site where DTGW is 49.5 m. AU-TTE is located near the eastern edge of the Ti Tree in plot 3 where DTGW is 8.8 m.

**2.3 Stable isotope analyses: deuterium, $\delta^{18}O$ and foliar $\Delta^{13}C$**

Branch xylem water was extracted by cryogenic vacuum distillation ( whereby samples are subject to a vacuum and water vapour is frozen using liquid nitrogen, as described in Ingraham and Shadel, 1992; West et al., 2006). Water from each branch sample was extracted for a minimum of 60-75 minutes (West et al., 2006). $\delta^2H$ and $\delta^{18}O$ analyses of branch water and groundwater were performed using a Picarro L2120-i Analyser for Isotopic $H_2O$. Five laboratory standards were calibrated against IAEA VSMOW2 – SLAP2 scale (Vienna Standard Mean Ocean Water 2, VSMOW2: $\delta^{18}O$ = 0 ‰ and $\delta^2H$ = 0 ‰; Standard Light Antarctic Precipitation 2, SLAP2: $\delta^{18}O$ = -55.5 ‰ and $\delta^2H$ = -427.5 ‰) and Greenland Ice Sheet Precipitation (GISP, $\delta^{18}O$ = -24.8 ‰ and $\delta^2H$ = -189.5 ‰) as quality control references (IAEA, 2009). The standard

deviation of the residuals between the VSMOW2 – SLAP2 value of the internal standards and the calculated values based on best linear fits was *ca* 0.2 ‰ for $\delta^{18}O$ and *ca* 1.0 ‰ for $\delta^2H$.

**2.4 Carbon isotope ratios of leaves**

Leaf samples stored in paper bags were completely dried in an oven at 60 ℃ for five days. After drying, each leaf sample was finely ground to powder with a Retsch MM300 bead grinding mill (Verder Group, Netherlands) until homogeneous. Between one and two milligrams of ground material was sub-sampled in 3.5 mm X 5 mm tin capsules for analysis of the stable carbon isotope ratio ($\delta^{13}C$), generating three representative independent values per tree. All $\delta^{13}C$ analyses were performed in a Picarro G2121-i Analyser (Picarro, Santa Clara, CA, USA) for isotopic $CO_2$. Atropine and acetanilide were used as laboratory standard references. Results were normalised with the international standards sucrose (IAEA-CH-6, $\delta^{13}C_{VPDB}$ = -10.45 ‰), cellulose (IAEA-CH-3, $\delta^{13}C_{VPDB}$ = -24.72 ‰) and graphite (USGS24, $\delta^{13}C_{VPDB}$ = -16.05 ‰). Standard deviation of the residuals between IAEA standards and calculated values of $\delta^{13}C$ based on best linear fit was *ca* 0.5 ‰.

**2.5 Calculation of WUE$_i$ from $\delta^{13}$C and hence $\Delta^{13}$C**

WUE$_i$ was determined from $^{13}$C discrimination ($\Delta^{13}$C), which is calculated from bulk-leaf carbon isotope ratio ($\delta^{13}$C), using the following equations (Werner et al., 2012):

$$\Delta^{13}C \ (‰) = \frac{R_a - R_p}{R_p} = \frac{\delta^{13}C_a - \delta^{13}C_p}{1 + \frac{\delta^{13}C_p}{1000}} \tag{1}$$

$$WUE_i = \frac{c_a(b - \Delta^{13}C)}{1.6(b - a)} \tag{2}$$

**2.6 Leaf vein density**

A small sub-section (approximately 1 cm$^2$) of all leaves sampled for $^{13}$C were used for LVD analysis, providing three leaf sub-sections per tree, nine samples per species. Due to the small size of *A. aneura* phyllodes, several phyllodes were combined and ground for $^{13}$C analysis and one whole phyllode was used for LVD analysis. Each leaf section was cleared and stained following the approach described in Gardner (1975). A 5% (w/v) NaOH solution was used as the principal clearing agent. Leaf-sections were immersed in the NaOH solution and placed in an oven at 40 ℃ overnight (Gardner, 1975).

Phyllodes of *Acacia* proved difficult to clear effectively and were kept in the oven longer than overnight to aid clearing. Once cleared, the partially translucent leaf sections were stained with a 1% (w/v) safranin solution. Most leaf sections were stained for up to three minutes and then soaked with a 95% (w/v) ethanol solution until the vein network was sufficiently stained and the majority of colour was removed from the lamina. After staining, the cuticle was removed to aid in identifying the vein network. Following cuticle removal, leaf sub-sections were photographed using a Nikon microscope (model: SMZ800) at 40 x magnification. Finally, minor veins were traced by hand and LVD was calculated as total vein length per unit area (mm/mm$^2$) using ImageJ version 1.48 (National Institutes of Health, USA).

**2.7 Data and statistical analysis**

Species-mean values (n = 9) for the dominant overstorey species at each location were calculated for $\delta^2$H, $\delta^{18}$O, $\delta^{13}$C and LVD. Relationships between $\Delta^{13}$C and WUE$_i$ with DTGW were tested using regression analysis after testing for non-normality (Shapiro-Wilk test, $\alpha$ = 0·05) and homogeneity of variances (Bartlett test). A Tukey's *post-hoc* test for multiple comparisons across sites was used to test for significance of variation as a function of DTGW. Breakpoints in functions with DTGW were determined using segmented regression analyses whereby the best fitting function is obtained by maximising the statistical coefficient of explanation. The least squares method was applied to each of the two segments while minimising the sum of squares of the differences between observed and calculated values of the dependent variables. Next, one-way ANOVA was applied to determine significance of regressions and breakpoint estimates within a given season. We fully acknowledge that the number of plots with shallow DTGW was sub-optimal, but constraints arising from species distribution across the TI-Tree precluded additional sampling.

**3 Results**

**3.1 Meteorological conditions during the study period**

Mean daily temperature and mean daily vapour pressure deficit (VPD) were largest in summer (December – February) and smallest in winter (June – August) (Figure 1). Daily sums of rainfall showed that the DTW 8.8 m and 49.4 m sites received 265 mm and 228 mm rainfall, respectively, between the September 2013 and April 2014 sampling dates, representing more than 73% and 60% of the total respective rainfall received from January 2013 to June 2014 at these sites.

**3.2 Variation in source-water uptake**

Xylem water isotope ratios for *A. aneura* were widely divergent from the bore water stable isotope ratios (Fig. 2a) in both wet and dry-seasons, indicative of lack of access to groundwater. In contrast, $\delta^2$H and $\delta^{18}$O of xylem water in *E. camaldulensis* and *C. opaca* were predominantly (with two exceptions) tightly clustered around $\delta^2$H and $\delta^{18}$O of groundwater in the bores located within 50 m of the trees (Fig. 2b). There was little variation in xylem water composition

between the end of the dry and end of the wet-seasons for either species, reflecting the consistent use of groundwater by these two species.

$\Delta^{13}C$ of *A. aneura*, sampled in the *Corymbia* savanna and Mulga plots was not significantly correlated with DTGW, in either season or across all values of DTGW (ANOVA F = 1.78; p > 0.05; Fig. 3a, b). As a consequence of these patterns in $\Delta^{13}C$, WUE$_i$ did not vary significantly for *A. aneura* across sites differing in DTGW (Fig. 4a) despite the large variability in WUE$_i$ (ranging 62 – 92 µmol mol$^{-1}$) across sites and seasons. By contrast, foliar $\Delta^{13}C$ of *E. camaldulensis* and *C. opaca* declined significantly with increasing DTGW in both seasons at the sites where DTGW was relatively shallow (DTGW < *ca* 12 m). Segmented regression analysis shown in Figure 3 (c) and (d) yielded breakpoints at 11.17 $\pm$ 0.54 m in September (ANOVA F = 11.548; df = 2, 38; P < 0.01) and 9.81 $\pm$ 0.4 m in April (ANOVA F = 14.67; df = 2, 47; P < 0.01), which represent the seasonal maximum depths from which groundwater can be extracted by these species. Thus, WUE$_i$ of *E. camaldulensis* and *C. opaca* was significantly smaller at the shallowest site than at sites with DTGW > 13.9 m (Fig. 4b) but did not differ significantly across the deeper DTGW range (13.9 – 49.5 m; Fig. 4b).

As distance from Allungra creek increased, foliar $\Delta^{13}C$ for both *E. camaldulensis* and *C. opaca* declined significantly (Fig. 5a), with concomitant increases in WUE$_i$ (Fig. 5b). There was no significant relationship for $\Delta^{13}C$ nor WUE$_i$ with distance from the creek for *A. aneura* (data not shown).

**3.3 Leaf vein density**

Leaf vein density (LVD) did not vary significantly with increasing DTGW for *A. aneura* (Fig. 6a), but a significant increase was observed for DTGW < *ca* 10 m in the two deep-rooted species (Fig. 6b). The break point for *E. camaldulensis* and *C. opaca* (Fig. 6b; 9.36 m $\pm$ 0.6 m; ANOVA F = 6.38; P<0.05) agreed well with the two previous estimates (cf Figs. 3 and 6), although again, constraints imposed by species distributions severely limited the number of samplings available at shallow DTGW sites.  As with LVD and DTGW, no relationship was observed between LVD and $\Delta^{13}C$ for *A. aneura* (Fig. 7a), but a significant linear decline in $\Delta^{13}C$ with increasing LVD was observed for *E. camaldulensis* and *C. opaca* (Fig. 7b).

**4 Discussion**

Analyses of stable isotopes of bore water (i.e groundwater) and xylem water across a DTGW gradient established that *A. aneura* adopted an "opportunistic" strategy of water use and was dependent on rainfall stored within the soil profile. This is consistent with previous studies, where Mulga was shown to be very responsive to changes in upper soil moisture content, as expected given their shallow rooting depth and the presence of a shallow (< 1.5 m) hardpan below stands of Mulga (Eamus et al., 2013; Pressland, 1975). Furthermore, very low predawn leaf-water-potentials (< -7.2 MPa; Eamus et al., 2016) and very high sapwood density (0.95 g cm$^{-3}$; Eamus et al., 2016) in *A. aneura* of the Ti Tree and which are strongly correlated with aridity, confirms that they rely on soil water without access to groundwater, consistent with the findings of Cleverly et

al.,. (2016b). By contrast, analyses of stable isotopes in groundwater and xylem water of *E. camaldulensis* and *C. opaca* established their access to groundwater, as has been inferred previously because of their large rates of transpiration in the dry-season and consistently high (close to zero) predawn water potentials (Howe et al., 2007; O'Grady et al., 2006a; O'Grady et al., 2006b). Importantly, we observed no significant change in xylem isotope composition for these two deep-rooted species between the end of the wet-season and the end of the dry-season, further evidence of year-round access to groundwater at the shallowest DTGW sites.

One specific aim of the present study was to determine whether discrimination against $^{13}$C ($\Delta^{13}$C) and resultant intrinsic water-use efficiency ($WUE_i$) could be used to identify access to groundwater. An increase in foliar $\Delta^{13}$C represents decreased access to water and increasing $WUE_i$ (Leffler and Evans, 1999; Zolfaghar et al., 2014; Zolfaghar et al., 2017). The shallow-rooted *A. aneura* did not show any significant relationship of $\Delta^{13}$C with DTGW during either season. Consequently mean $WUE_i$ showed no significant trend with increasing DTGW, consistent with the conclusion that *A. aneura* only accessed soil water during either season. In contrast, $\Delta^{13}$C of groundwater-dependent vegetation (*E. camaldulensis* and *C. opaca*) declined significantly along the DTGW gradient to a threshold value of DTGW of between 9.4 m (derived from LVD results) and 11.2 m (derived from $\Delta^{13}$C analyses in Sept). Where DTGW was below the threshold (DTGW > *ca* 12m), $\Delta^{13}$C became independent of DTGW. Consequently, $WUE_i$ increased significantly in *E. camaldulensis* and *C. opaca* as DTGW increased from 4.4 m to these thresholds (p < 0.001) but did not vary with further increases in DTGW. We therefore conclude that foliar $\Delta^{13}$C (or $WUE_i$) can be used as an indicator of groundwater access by vegetation. $\Delta^{13}$C is less expensive and easier to measure than stable isotope ratios of water ($\delta^2$H and $\delta^{18}$O) in groundwater, soil water and xylem water. Furthermore, canopies are generally more accessible than groundwater. Globally, identification of groundwater-dependent ecosystems has been hindered by the lack of a relatively cheap and easy methodology (Eamus et al., 2015), thus $\Delta^{13}$C shows great promise for identifying groundwater-dependent vegetation and ecosystems.

Whilst acknowledging the sub-optimal distribution of samples within the shallow DTGW range (< 10 m) from which break-points in regressions were calculated (Figs. 3, 6), which arose because of the natural distribution of trees across the basin, we can ask the question: are the speculated depths beyond which groundwater appears to become inaccessible supported by other independent studies of Australian trees? Eamus et al., (2015) present the results of a seven site (seven sites across the range 2.4 – 37.5 m DTGW), 18 trait, 5 species study and identify a breakpoint between 7 – 9 m. Similarly, two recent reviews identify lower limits to root extraction of groundwater of 7.5 m (Benyon et al., 2006) and 8-10 m (O'Grady et al., 2010) while Cook et al., (1998b) established a limit of 8 – 9 m for a Eucalypt savanna. We therefore conclude that our estimates to limit of groundwater accessibility appear reasonable.

Figure 8 shows combined $\Delta^{13}$C from four Australian studies (Miller et al., 2001; Stewart et al., 1995; Taylor, 2008; Rumman et al., 2017), including one continental-scale study of foliar $\Delta^{13}$C (Rumman et al., 2017). A single regression describes the

data of all four independent studies. Thus, when rainfall is the sole source of water for vegetation, $\Delta^{13}C$ is strongly correlated with annual rainfall. In contrast, the mean $\Delta^{13}C$ for *E. camaldulensis* and *C. opaca* do not conform to the regression (Fig. 8). It appears that *E. camaldulensis* in the Ti Tree "behaves" as though it were receiving approximately 1700 mm of rainfall, despite growing at a semi-arid site (*ca* 320 mm average annual rainfall). This represents the upper limit to groundwater use by this species, assuming a zero contribution from rainfall (which is clearly very unlikely). The upper limit to annual groundwater use for *C. opaca* was similarly estimated to be 837 mm (Fig. 8). If all of the water from rainfall is used by these two species (which is also very unlikely) then the lower limit to groundwater use is the difference between rainfall and the estimates derived from Figure 8, about 1380 mm for *E. camaldulensis* and 517 mm for *C. opaca*.

There any independent estimates of annual tree water-use for these two species which provide a valuable comparison to the estimates made above. O'Grady et al., (2009) showed that annual water use by riparian *E. camaldulensis* in the Ti-Tree was approximately 1642.5 $m^3$ $m^{-2}$ sapwood $y^{-1}$. Assuming average tree radius is 20 cm, sapwood depth is 2 cm and average canopy coverage of the ground is 25 $m^2$ per tree, this yields an annual water use of 1568 mm $y^{-1}$, encouragingly close to the estimate (1700 mm) derived from Figure 8. The estimate for annual water use by *C. opaca* from O'Grady et al (2009) is 837 mm, in reasonable agreement with the estimate from the average $\Delta^{13}C$ of *C. opaca* and the regression in Figure 8 (*ca* 900 mm). Because depth-to-groundwater for *C. opaca* is significantly larger (*ca* 8 -10 m) than that for *E. camaldulensis* (*ca* 2 - 4 m), the resistance to water flow imposed by the xylem's path length is larger for *C. opaca* than *E. camaldulensis* and therefore water-use may be expected to be smaller in the former than the latter, as observed.

Using an entirely different methodology from that used here, O'Grady et al., (2006c) estimated annual groundwater use by riparian vegetation on the Daly River in northern Australia to be between 694 mm and 876 mm, while O'Grady and Holland, (2010) showed annual groundwater use to range from 2 mm to > 700 mm in their continent-wide review of Australian vegetation. Therefore our estimates based on $\Delta^{13}C$ appear reasonable. We conclude that (a) *E. camaldulensis* and *C. opaca* are accessing groundwater (because annual water use greatly exceeded annual rainfall); (b) $\Delta^{13}C$ can be used as an indicator of groundwater use by vegetation; and (c) $\Delta^{13}C$ can provide estimates of upper and lower bounds for the rate of ground water use by vegetation.

## 4.1 Patterns of carbon isotope discrimination and intrinsic water-use efficiency along Allungra creek transects

For the two deep-rooted species, a significant decline in $\Delta^{13}C$ (and hence an increase in WUE$_i$) was observed with increasing distance from the creek (Fig. 6a, b). In contrast to results of O'Grady et al., (2006c) for a steeply rising topography in the Daly River, this is unlikely to be attributable to increased elevation since the change in elevation was minimal across each entire transect (< 3 m), and even smaller near Allungra Creek where most of the change in $\Delta^{13}C$ was recorded. Therefore the cause of the change in $\Delta^{13}C$ with distance from the creek was most likely to be a function of the frequency with which trees receive flood water and hence the amount of recharge into the soil profile (Ehleringer and Cooper, 1988; Thorburn et al.,

1994; Villeneuve et al., 2015; Singer et al., 2014). We therefore further conclude that foliar $\Delta^{13}C$ can be used as an indicator of access to additional water to that of rainfall, regardless of the source of that additional water (e.g., groundwater, flood recharged soil water storage, irrigation).

**4.2 Leaf vein density across the depth-to-groundwater gradient**

In our study, LVD was independent of DTGW for *A. aneura* but increased with DTGW for *E. camaldulensis* and *C. opaca* to a break point of 9.4 m, above which LVD was independent of DTGW. Increased DTGW reflects a declining availability of water resources (Zolfaghar et al., 2015), especially in arid-zones. Uhl and Mosbrugger (1999) concluded that water availability is the most important factor determining LVD. Sack and Scoffoni (2013) also showed LVD to be negatively correlated with mean annual precipitation in a 796 species meta-analysis. Therefore increasing LVD with increasing DTGW

is consistent with the positive effect of water supply on LVD, despite similar amounts of rainfall being received along the DTGW gradient.

Both *A aneura* and *C. opaca* in the present study showed LVDs close to the higher end of the global spectrum (Sack and Scoffoni, 2013), consistent with larger LVDs observed in "semi-desert" species. Higher LVDs allow for a more even spatial distribution of water across the phyllode or lamina during water-stress, which contributes to a greater consistency of

mesophyll hydration in species of arid and semi-arid regions (Sommerville et al., 2012). In turn, this allows continued photosynthetic carbon assimilation during water-stress (Sommerville et al., 2010). Presumably, a large LVD also decreases the resistance to water flow from minor veins to mesophyll cells, which is likely to be beneficial for leaf hydration as water availability declines while also facilitating rapid rehydration following rain in these arid-zone species. Large LVDs for *A. aneura* of semi-arid regions in Australia have been associated with rapid up-regulation of phyllode function with the return

of precipitation following drought (Sommerville et al., 2010), and such rapid up-regulation is crucial for vegetation in regions with unpredictable and pulsed rainfall like the Ti Tree (Byrne et al., 2008; Grigg et al., 2010).

LVD was negatively correlated with bulk-leaf $\Delta^{13}C$ (and thus positively correlated with $WUE_i$) in *E. camaldulensis* and *C. opaca*. What mechanism can explain the significant relationship observed between a structural leaf-trait (LVD) and a functional trait ($WUE_i$)? The stomatal optimization model (Medlyn et al., 2011) is based on the fact that transpiration (E)

and $CO_2$ assimilation (A) are linked via stomatal function. In order to gain carbon most economically while minimising water loss (i.e., optimization of the ratio A/E), stomata should function such that the marginal water cost of carbon assimilation $(\frac{\partial A}{\partial E})$ remains constant (Cowan and Farquhar, 1977; Farquhar and Sharkey, 1982). This aspect of stomatal control couples the structural traits involved with water-flow with traits associated with primary production (Brodribb and Holbrook, 2007) and explains observed correlations between $K_{leaf}$ and $A_{max}$ in a number of studies (Brodribb et al., 2007; Brodribb et

al., 2010; Brodribb et al., 2005; Brodribb and Jordan, 2008; Sack and Holbrook, 2006; Sack and Scoffoni, 2013). The length

of hydraulic pathway is directly proportional to $K_{leaf}$ (Brodribb et al., 2007) and the $A/g_s$ ratio determines foliar $\Delta^{13}C$ (and $WUE_i$) signatures in leaves. Thus, the constraint on $K_{leaf}$ by LVD affects the coordination between the processes of A and E and thereby might explain significant relationships between structural (LVD) and functional traits ($WUE_i$). For the two deep-rooted species, having access to groundwater resulted in convergence to a common solution for optimizing water supply
through veins with respect to E (Brodribb and Holbrook, 2007).

Several robust and testable predictions arise from the conclusion that *E. camaldulensis* and *C. opaca* are functioning at a semi-arid site as though they have access to *ca* 1700 or 900 mm rainfall, respectively. Species growing in high rainfall zones possess a suite of traits, including low density sapwood, large diameter xylem vessels, small resistance to vessel implosion, large SLA and a large maximum stomatal conductance, compared to species growing in arid regions (O'Grady et al., 2006b;
O'Grady et al., 2009; Wright et al., 2004). Therefore, these attributes should be present in *E. camaldulensis* and *C. opaca* if they are functioning as though they are growing in a mesic environment. We have previously established (Eamus et al., 2016; Santini et al., 2016) that these predictions are confirmed by field data and therefore conclude that analyses of foliar $\Delta^{13}C$ has global application to the preservation and understanding of GDEs.

## 4.3 Conclusions

We posed five questions regarding depth-to-groundwater (DTGW), foliar discrimination against $^{13}C$ ($\Delta^{13}C$) and leaf vein density (LVD) as underpinning the rationale for this study. We confirmed that access to shallow groundwater by *E. camaldulensis* and *C. opaca* (DTGW *ca* 0 −11 m) resulted in smaller $WUE_i$ than in *A. aneura*. We also demonstrated that LVD correlated with DTGW for the shallower depths (< 10 m) in *E. camaldulensis* and *C. opaca*, but not in *A. aneura*. We further demonstrated that there was correlation between LVD and $\Delta^{13}C$ (and hence $WUE_i$) for *E. camaldulensis* and *C.*
*opaca*, but not in *A. aneura*. Similarly, as distance increased from a creek near a flood-out associated with aquifer recharge, foliar $\Delta^{13}C$ increased and $WUE_i$ decreased for *E. camaldulensis* and *C. opaca,* but not for *A. aneura*. Finally, we conclude that foliar $\Delta^{13}C$ can be used as an indicator of utilisation of groundwater or stored soil water by vegetation in arid regions, providing an inexpensive and rapid alternative to the stable isotopes of water that have been used in many previous studies. The observation that the $\Delta^{13}C$ of the two groundwater-using species was distant from the continental regression of $\Delta^{13}C$
against rainfall (Fig. 8) is strong evidence of the value of $\Delta^{13}C$ as an indicator of utilisation of water that is additional to rainfall, and this supplemental water can be derived from either groundwater or soil recharge arising in flood-out zones of creeks.

**5 Author contributions**

All authors contributed to field data sampling. RR undertook all the isotope analyses and statistical analyses and wrote the thesis that formed the basis of this Ms. DE oversaw the design and implementation of the entire project. All authors contributed to writing this Ms and interpretation of the data.

**6 Acknowledgements**

The authors would like to acknowledge the financial support of the Australian Research Council for a Discovery grant awarded to DE.

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

Figures

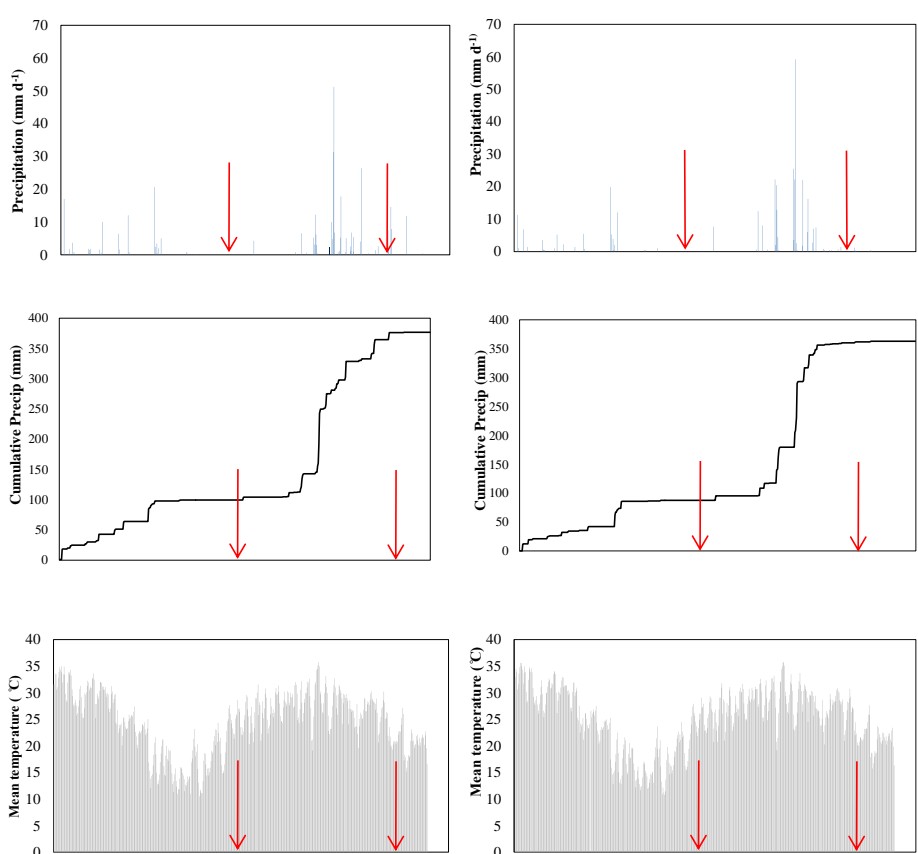

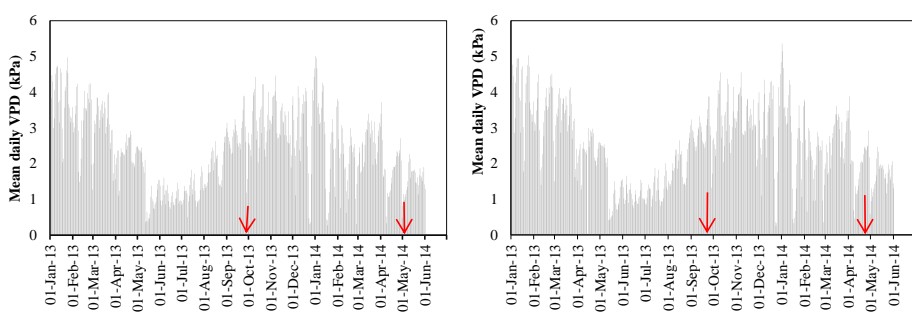

Figure 1:     Mean daily meteorological conditions of daily precipitation, cumulative precipitation, mean air
temperature and vapour pressure deficit Jan 2013 - Jun 2014. Red lines show sampling periods
in September 2013 (late dry-season) and April 2014 (late wet-season). Left panels show data
from the western EC tower (DTGW 49.4 m), and right panels show data for the eastern EC tower
(DTGW 8.8 m).

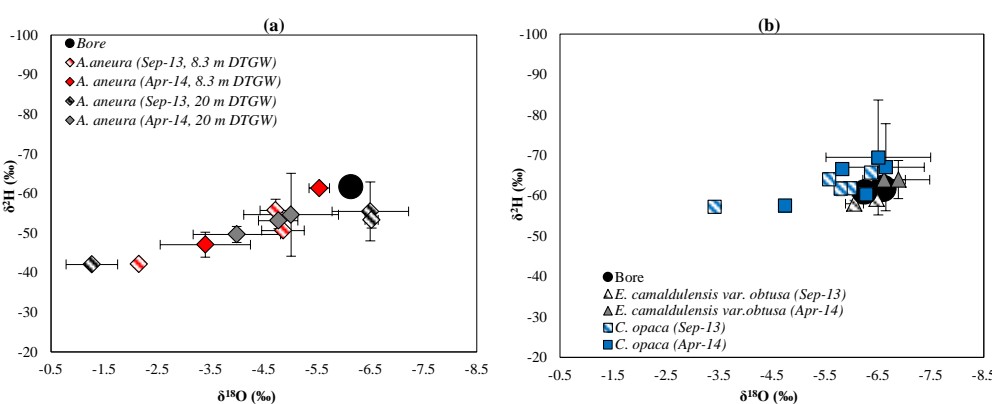

Figure 2:    Comparison of xylem and bore water δ²H-δ¹⁸O plots for *Acacia aneura* sampled from 8.3 m and 20 m DTGW (a) and *Eucalyptus camaldulensis* and *Corymbia opaca* sampled from 4.4 m, 8.8 m and 13.9 m DTGW (b). Error bars represent ± one standard error.

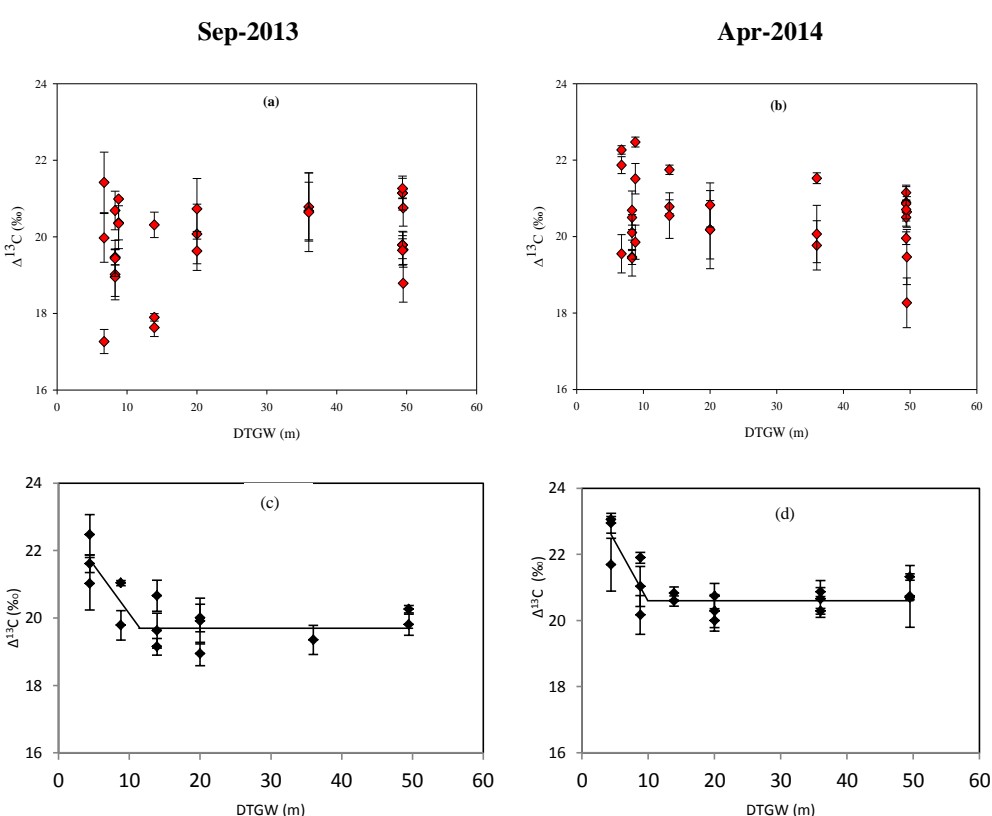

Figure 3: Carbon isotope discrimination in leaf dry matter ($\Delta^{13}C$) plotted as a function of depth-to-groundwater (DTGW) in the Ti Tree basin. Red points in panel (a) and (b) for *A. aneura* and black points in panel (c) and (d) *C. opaca* and *E. camaldulensis*. Left and right panels show Sep-2013 and Apr-2014 sampling respectively. Lines in panels (c) and (d) are from segmented regression. Error bars represent ± one standard error.

**(a)**

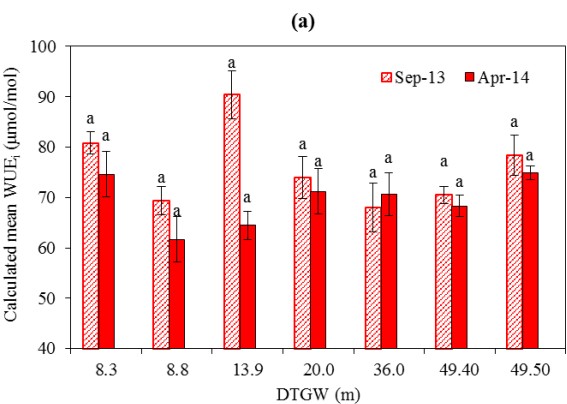

**(b)**

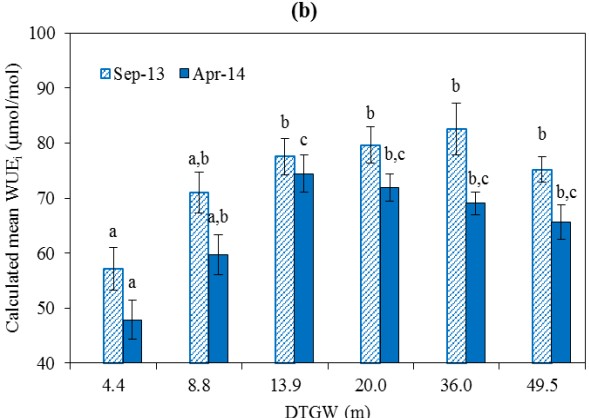

Figure 4: Leaf intrinsic water-use-efficiency ($WUE_i$) calculated from $\Delta^{13}C$ in shallow-rooted *A. aneura* (a) and deep-rooted *E. camaldulensis* and *C. opaca* (b) across study sites for Sep-2013 (patterned column) and Apr-2014 (filled column). Bars within a season with the same letter are not significantly different across the depth-to-groundwater gradient (Tukey HSD, $p < 0.05$). Error bars represent ± one standard error.

(a)

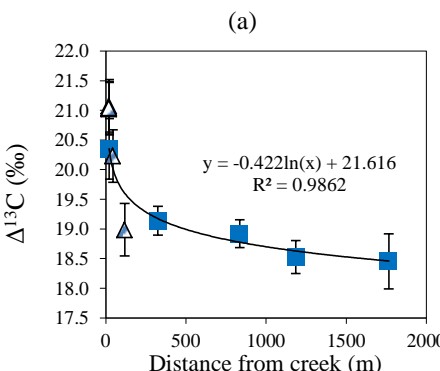

(b)

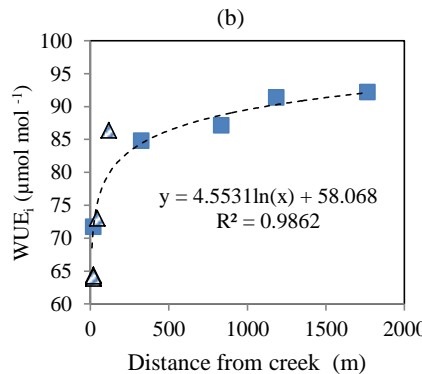

Figure 5: $\Delta^{13}C$ (a) and $WUE_i$ (b) of deep-rooted species sampled across Ti Tree basin plotted as functions of distance from Allungra Creek bed. Striped triangles represent *E. camaldulensis* and blue squares represent *C. opaca*. Error bars represent ± one standard error. The regression is fitted only to the *E. camuldulensis* data. Note that the largest value of $\Delta^{13}C$ and the lowest value of WUE for *C. opaca* are three overlapping samples.

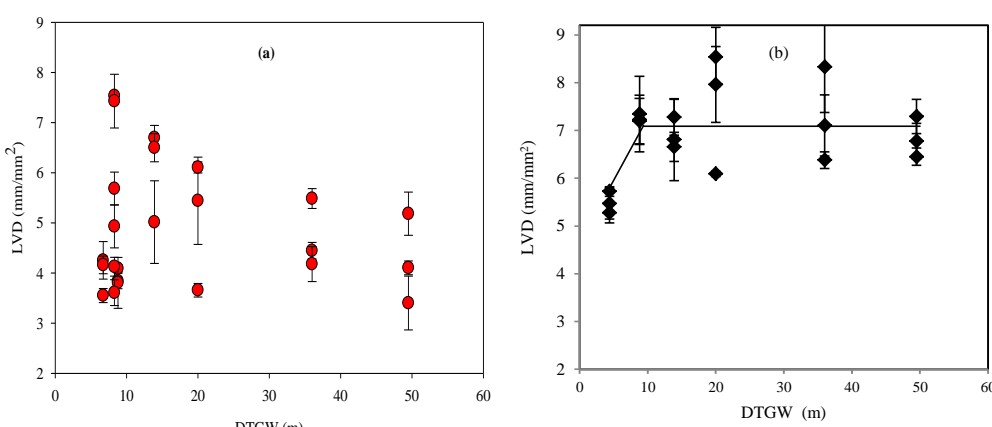

Figure 6:     Leaf vein density (LVD) of *A. aneura* (panel a) or *E. camaldulensis* and *C. opaca* (panel b) as a
              function of depth-to-groundwater. Each symbol represents mean LVD calculated from three
              individual leaves. Error bars represent ± one standard error. A statistically significant correlation
              derived from segmented linear regression of leaf vein density, for *E. camaldulensis* and *C. opaca*
              with depth-to-groundwater (DTGW) is shown in panel (b). The $r^2$ and standard deviation slope of
              the regression below the break point in (b) are 0.976 a d 0.0031 respectively.

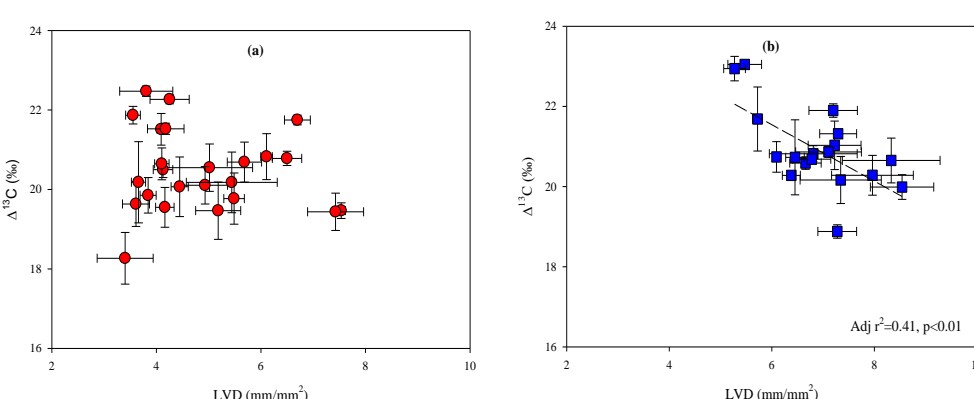

Figure 7:    Relationships of leaf vein density of (a) *A. aneura* and (b) *E. camaldulensis* and *C. opaca* with bulk-leaf $\Delta^{13}C$. Each symbol represents mean LVD and $\Delta^{13}C$, with both variables measured on the same leaf. Error bars represent ± one standard error. A statistically significant correlation of LVD and $\Delta^{13}C$ of *E. camaldulensis* and *C. opaca* is plotted with a dashed line.

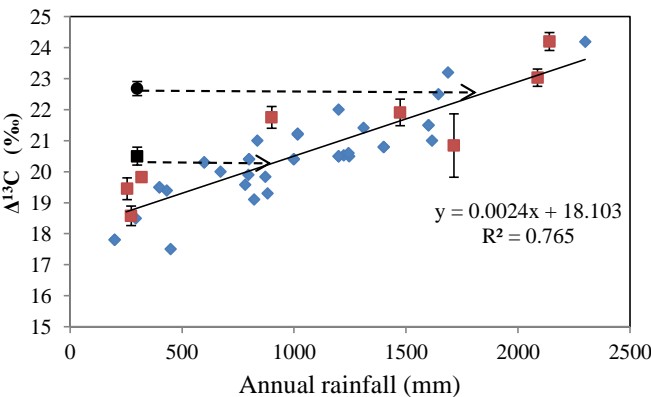

Figure 8:     Relationships of discrimination against carbon-13 ($\Delta^{13}C$) with annual rainfall observed in
different studies across Australia. The diamonds represent observations made in eastern Australia
(Stewart et al., 1995), northern Australia (Miller et al., 2001) and sites in New South Wales
(Taylor, 2008). The red squares are data from a continental-scale assessment of foliar $\Delta^{13}C$
10        (Rumman et al., 2017). The black circle is the mean $\Delta^{13}C$ of *E. camaldulensis* and the black
square is the mean $\Delta^{13}C$ *C. opaca*, both of which were measured in this study. The 95 % CI for
the mean $\Delta^{13}C$ is $\pm$ 0.403 and the s.e of the slope is 0.000231.  The black arrows indicate the
rainfall that would be required to account for the $\Delta^{13}C$ for *E. camaldulensis* and *C. opaca* if
these two species relied only upon rainfall.

# Speculations on the application of foliar $^{13}$C discrimination to reveal groundwater dependency of vegetation, provide estimates of root depth and rates of groundwater use

Rizwana Rumman[1], James Cleverly[1], Rachael H. Nolan[1], Tonantzin Tarin[1] and Derek Eamus[1,2]

[1]Terrestrial Ecohydrology Research Group

School of Life Sciences

University of Technology Sydney

PO Box 123

Broadway

NSW 2007

Australia

*Correspondence to*: Derek Eamus  (Derek.Eamus@uts.edu.au)

**Abstract.**

Groundwater-dependent vegetation is globally distributed, having important ecological, social and economic value. Along with the groundwater resources upon which it depends, this vegetation is under increasing threat through excessive rates of groundwater extraction.

In this study we examined one shallow-rooted and two deep-rooted tree species at multiple sites along a naturally occurring gradient in depth-to-groundwater. We measured (i) stable isotope ratios of leaves ($\delta^{13}$C), xylem and groundwater ($\delta^{2}$H and $\delta^{18}$O); and (ii) leaf vein density. We established that foliar discrimination of $^{13}$C ($\Delta^{13}$C) is a reliable indicator of groundwater use by vegetation and can also be used to estimate rooting depth. Through comparison with a continental-scale assessment of foliar $\Delta^{13}$C, we also estimated the upper limits to annual rates of groundwater use. We conclude that maximum rooting depth for both deep-rooted species ranged between 9.4 m and 11.2 m and that annual rates of groundwater use ranged *ca* 1400 – 1700 mm for *Eucalyptus camaldulensis* and 600 – 900 mm for *Corymbia opaca*. Several predictions about hydraulic and leaf traits arising from the conclusion that these two species made extensive use of groundwater were supported by additional studies of these species in central Australia.

**1 Introduction**

Drylands cover 41% of the earth's total land area (Reynolds et al., 2007) and are sub-categorized as hyper-arid, arid, semi-arid and dry sub-humid areas. Hyper-arid, arid and semi-arid regions are characterized by chronic water shortage with unpredictable rainfall (Clarke, 1991). Approximately 40 % of the world's population reside in drylands and groundwater
represents a major water resource both for human consumptive use but also for groundwater dependent ecosystems (GDEs' Eamus et al., 2005). Sustainable management of both groundwater and GDEs requires identification of the location of GDEs, rooting depth of vegetation and rates of groundwater use but attaining such information present significant technical and cost challenges (Eamus et al., 2015).

Approximately 70% of Australia is classified as semi-arid or arid (Eamus et al., 2006; O'Grady et al., 2011). Furthermore,
annual potential evaporation exceeds annual rainfall across most of the continent, thus most Australian biomes are water-limited according to the Budyko (1974) framework (Donohue et al., 2009). On average, central Australia receives less than 350 mm y$^{-1}$ of rainfall, making water a primary limiting resource (Eamus et al., 2006). Because surface water bodies in this region are mostly ephemeral (NRETAS, 2009, although see Box et al., 2008, regarding the small number of permanent water bodies), groundwater plays an important role in maintaining ecosystem structure and function of terrestrial (especially
riparian) vegetation (Eamus et al., 2006). Owing to the remoteness of much of Australia's interior, few studies have investigated groundwater use by vegetation communities in these semi-arid regions.

Stomatal conductance is regulated to maximise carbon gain whilst simultaneously minimising transpiration (Cowan and Farquhar, 1977; Medlyn et al., 2011) and is sensitive to both soil and atmospheric water content (Prior et al., 1997; Thomas and Eamus, 1999). Intrinsic water-use efficiency (WUE$_i$), defined by the ratio of carbon gain to stomatal conductance)
provides valuable insights on how vegetation responds to variation in water availability (Beer et al., 2009). Declining water supply results in increased WUE$_i$ as stomatal conductance declines (Eamus et al., 2013). Discrimination against the $^{13}$C isotope ($\Delta^{13}$C) is commonly used to calculate WUE$_i$. $\Delta^{13}$C provides a time-integrated measure of WUE$_i$ (Cernusak et al., 2011); in this study we examined spatial and seasonal patterns in $\Delta^{13}$C across three tree species in the Ti Tree basin.

The present study was undertaken in the Ti Tree basin, which is the location of an important groundwater resource in central
Australia (Cook et al., 2008a). Rainfall occurs mostly in large events during the austral summer (Dec – Mar); thus there is minimal rainfall available for vegetation use over prolonged periods. The dry-season in this region is characterized by declining soil water availability and high vapour pressure deficits (Eamus et al., 2013). Previous studies have documented several surprising attributes for a number of tree species in the Ti Tree. O'Grady et al.,. (2009) observed that, despite living in an extremely water limited environment, the specific leaf-area (SLA) of *Corymbia opaca* and *Eucalytptus camaldulensis*
were similar more to those from highly mesic environments than to species from arid environments. Similarly, Santini et al.,. (2016) observed that xylem wall thickness and vessel implosion resistance were significantly smaller in *E. camaldulensis*

and *C. opaca* than in shallow-rooted *Acacia aneura*. Finally, rates of water use and changes in midday water potential between the end of the wet-season and the end of the dry-season were minimal for *E. camaldulensis* and *C. opaca* but were very large for *Acacia aptaneura* (which was previously classified as *Acacia aneura*; Maslin and Reid, 2012; Nolan et al., 2017). *A. aneura* and *A. aptaneura* often intermix with other members of the large Mulga complex of closely related *Acacia* species (Wright et al., 2016), thus we will refer *Acacia* spp. in the Mulga complex by the primary type, *A. aneura*. *A. aneura* is shallow-rooted and associated with shallow hard pans in this catchment (*ca* 1 m below ground surface; Cleverly et al., 2016a; Cleverly et al., 2016b) which prevents access to the groundwater below. *E. camaldulensis* is a riparian species, confined to narrow corridors along the ephemeral streams in the Ti Tree (the Woodforde River and Allungra Creek) where groundwater depth is shallow (< 3m); *C. opaca* is deep- rooted and may access groundwater to depths of 8 m or more (O'Grady et al., 2009). These observations lead to the first hypothesis tested in the present study: that $WUE_i$ of *A. aneura* would be significantly larger than that of *E. camaldulensis* and *C. opaca* because reliance on shallow stores of water by Mulga imposes severe restrictions on water use, thus resulting in a large $WUE_i$.

Vertical and horizontal distance from rivers receiving groundwater inflows in arid-zones influences the degree to which trees access groundwater (O'Grady et al., 2006a; Thorburn et al., 1994). Trees closest to the river (vertically and horizontally) have xylem water deuterium and $^{18}O$ isotope ratios ($\delta^2H$ and $\delta^{18}O$, respectively) that are close to the ratios of river and groundwater; trees further from the river have xylem water ratios increasingly different from those of groundwater and the river (O'Grady et al., 2006a). In endoreic basins like Ti Tree, evaporation of near-surface soil water imposes additional fractionation of $\delta^2H$ and $\delta^{18}O$ (Craig, 1961) relative to groundwater, thus $\delta^2H$ and $\delta^{18}O$ in xylem provides information on plant water source and climate (Cullen and Grierson, 2007). Variation in plant water sources with distance from the river can affect stomatal conductance and $WUE_i$. In this study we tested the hypothesis that $WUE_i$ would increase with distance from the creek.

Differential access to water among co-occurring species within a biome results in variation of several morphological traits, including SLA (Warren et al., 2005), Huber value (Eamus et al., 2000; Sperry, 2000) and wood density (Bucci et al., 2004; Hacke et al., 2000). Leaf vein density (LVD) is a trait that influences whole-plant performance. From a resource investment perspective, leaves are composed primarily of two components: mesophyll that undertakes photosynthesis and a leaf-vein network which delivers water and nutrients to the leaf. Investment in leaf veins is underpinned by resource allocation strategies (Niinemets et al., 2007; Niinemets et al., 2006; Niklas et al., 2007). Leaf-vein density is responsive to several environmental variables, but especially aridity (Uhl and Mosbrugger, 1999). Furthermore, LVD is positively correlated with leaf hydraulic conductance ($K_{leaf}$), maximum photosynthetic rate and leaf-level gas-exchange rates (Brodribb et al., 2007; Sack et al., 2003; Sack and Frole, 2006; Sack and Holbrook, 2006). ~~One aim of the present study was to~~Consequently we investigate whether investment in LVD of three dominant overstorey tree species was affected by increasing depth-to-groundwater (DTGW~~.~~).

The propensity for leaves to lose water matches the capacity of xylem to deliver the same volume of water (Brodribb and Holbrook, 2007; Meinzer and Grantz, 1990; Sperry, 2000), and positive correlations consequently occur between leaf-hydraulic conductance ($K_{leaf}$) and LVD (Brodribb et al., 2007; Sack and Holbrook, 2006). LVD provides a direct estimate of $K_{leaf}$ because it correlates with the distance water must traverse from termini of the xylem-network to sites of evaporation

(Brodribb et al., 2010). Since transpiration is directly linked to availability of water to roots, we hypothesised that LVD will be correlated with depth-to-groundwater (DTGW) in plants for which groundwater is accessible; this correlation should be absent in species with shallow roots which cannot access groundwater. Whilst a number of studies have demonstrated increased LVD with increasing aridity along rainfall gradients (Brodribb et al., 2010; Brodribb and Holbrook, 2003; Sack and Holbrook, 2006), this relationship has not, to our knowledge, been examined in relation to DTGW. Finally, because

LVD is strongly correlated with $K_{leaf}$ and rates of leaf-scale gas exchange (Brodribb et al., 2007; Sack et al., 2003; Sack and Frole, 2006), we hypothesise that LVD will be significantly correlated with $\Delta^{13}C$ (and hence $WUE_i$).

To summarize, we address the following questions:

1. Does access to groundwater by *E. camaldulensis* and *C. opaca* result in significantly smaller $WUE_i$ compared to *A. aneura* (Mulga)?

2. Does LVD correlate with DTGW in the three species examined?

3. Is there a correlation between LVD and $\Delta^{13}C$ (and hence $WUE_i$) for the three species examined?

4. Does horizontal and vertical distance from a known river flood-out zone influence foliar $\Delta^{13}C$ and $WUE_i$ of co-occurring species?

5. Can foliar $\Delta^{13}C$ be used as an indicator of utilisation of groundwater by vegetation of arid regions?

6. Can foliar $\Delta^{13}C$ be used to estimate rooting depth and upper and lower bounds of rates of groundwater use?

**2 Materials and methods**

**2.1 Site description**

The study was conducted in the Ti Tree Basin, a 5500 km$^2$ basin located approximately 200 km north of Alice Springs NT

and 180 km north of the Tropic of Capricorn (22.28°S, 1933.25°E, 549 m asl). Climate is characterized as tropical and arid with hot summers and warm winters. The nearest Bureau of Meteorology station (Territory Grape Farm; Met Station 015643; within 25 km of all study sites) recorded mean and median annual precipitation of 319.9 and 299 mm, respectively (1987 - May 2016; http://www.bom.gov.au/). Of the annual median rainfall, 72% falls during the summer months (December - February) and 86% falls during the monsoon season (November - April). Mean minimum and maximum

monthly temperatures range from 5°C and 22.6°C in July to 22°C and 37.5°C in January.

The soil is a "red kandosol" (74:11:15 sand: silt: clay; Eamus et al., 2013), typical of large portions of semi-arid Australia, and has a high potential for drainage (Morton et al., 2011; Schmidt et al., 2010). Patches of hard siliceous soil is often observed and are likely surface expressions of the underlying hardpan (Cleverly et al., 2013), a common formation in the top 1 – 1.5 m in this type of soil (Cleverly et al., 2013; Cleverly et al., 2016a; Cleverly et al., 2016b; Morton et al., 2011). The

major potable source of water for this region is a large underground reservoir, recharged mainly by seepage from creek/river channels and their flood-out zones, "mountain" front recharge and by occasional very heavy rainfall events (NRETAS, 2009; Calf et al., 1991). The Ti Tree basin has a natural gradient in DTGW. The depth of the water table below ground level is shallow (< 2m) in the northern part and groundwater is lost through evapotranspiration (Shanafield et al., 2015), whereas DTGW reaches 60 m in the southern and western part of the basin and 20 – 40 m in the eastern region (NRETA, 2007).

All study sites were characterized as being in one of three distinct vegetation types (Nolan et al., 2017; Cleverly et al., 2016a): (1) riparian, predominantly consisting of *Eucalyptus camaldulensis var. obtusa*, which line the banks of the ephemeral streams in the Ti Tree basin (Woodforde River and Allungra Creek); (2) low mixed Mulga woodland (*A. aneura* F.Muell. ex Benth., *A. aptaneura* Maslin & J.E.Reid, *A. kempeana* F.Muell.) with an understorey of shrubs, herbs and $C_3$ and $C_4$ grasses; (3) tall, open *Corymbia* savanna with extensive Spinifex grass (*Triodia spp.*), sparse *Corymbia opaca* (D.J.Carr

& S.G.M.Carr) K.D.Hill & L.A.S.Johnson trees and occasional Mulga trees. The Woodforde River and Allungra Creek are ephemeral streams that flow only after large extensive rainfall events. Nonetheless, perched aquifers beneath their riparian corridors are recharged by large storms (Villeneuve et al., 2015), providing long-term access to groundwater near ephemeral streams. Allungra Creek and its flood-out zone represent zones of local recharge (NRETAS 2009). Overbank flooding and sheet flow occur in flood-outs where the Woodforde River and Allungra Creek enter the basin and split into a network of

smaller channels (NRETA, 2007), resulting in an estimated 1.8 ML of groundwater recharge per year (NRETAS, 2009).

Four sampling plots were established for determination of foliar $\Delta^{13}C$ and of $\delta^{18}O$ and $\delta^2H$ values for xylem water and groundwater from a nearby bore. DTGW in each of the four plots was 4.4 m, 8.3 m, 8.8 m and 13.9 m, respectively. One of the four plots was located on the banks of the Woodforde River (plot 1, DTGW = 4.4 m). *E. camaldulensis* is the dominant tree species in plot 1, and *C. opaca* is also present. In the second plot, *A. aneura* is the dominant species (plot 2, DTGW =

8.3 m). *C. opaca* is the dominant tree species in the two remaining plots (plot 3, DTGW = 8.8 m; plot 4, 13.9 m).

$\Delta^{13}C$ of *E. camaldulensis*, *C. opaca* and *A. aneura* were examined along three additional transects to investigate the influence topography and of distance from a creek on $WUE_i$. The three transects established perpendicular to the banks of Allungra Creek, which is in an area of known groundwater recharge at the base of the hills that bound the southern extent of the basin (NRETAS, 2009). One of the three transects was located near a permanent waterhole near the flood-out and the

bottom of Allungra Creek. Transects two and three were 1 – 2 km upstream of the water hole. Transects extended from the shore within 1 m of the creek up to a maximum of 1800 m from the creek (vertical distance from the creek bed ranged 0 – 4

m; Supplementary Fig S2a –c), across which vegetation graded from riparian forest to *Corymbia* open savanna, with occasional Mulga trees interspersed throughout.

In addition to the four plots and three transects, "spot sampling" for foliar $\Delta^{13}C$ alone was performed at ~~2 3 further additional~~three sites to extend the examination of variation in WUE$_i$ ~~as a function of DTGW:~~to 20 m, 36 m and 49.5 m DTGW for *A. aneura* or 20 m and 36 m for *C. opaca*. Finally, continental sampling of foliar $\Delta^{13}C$ of multiple dominant tree species was undertaken at seven sites distributed across Australia (Rumman et al., 2017) (Table S1; Karan et al., 2016; Rumman 2017) in order to allow comparison of foliar $\Delta^{13}C$ of our three Ti-Tree species with a continental-scale regression of $\Delta^{13}C$ with rainfall.

**2.2 Leaf sampling protocols and meteorology**

Sampling was undertaken in April 2014 (end of wet-season) and September 2013 (end of dry-season). Three mature, healthy leaves on each of three branches from two or three replicate trees were sampled for $\Delta^{13}C$ in all plots, transects and spot sampling sites. In addition, terminal branches of the trees in the four plots were collected for deuterium and $\delta^{18}O$ analyses of xylem water. ~~Bore water samples were also collected in the four plots and 2 3 spot sampling sites.~~Bore water samples were also collected in the four plots and the three spot sampling sites using a Groundwater samples were also collected from the bores located at each site using HydraSleeve no-purge groundwater sampler (Cordry, 2003). Sampling along the three transects occurred at three or four points along each transect. The three trees of each of the dominant species at each location were located within 50 m of the bore. Leaves for leaf vein analysis (see below) were collected during Sept 2013.

Climate conditions preceding and during the sampling periods were obtained from two eddy-covariance towers (Fluxnet sites AU-ASM and AU-TTE; Cleverly et al., 2016a; Cleverly 2011; Cleverly 2013). AU-ASM is located to the west of the Ti Tree at the spot sampling site where DTGW is 49.5 m. AU-TTE is located near the eastern edge of the Ti Tree in plot 3 where DTGW is 8.8 m.

**2.3 Stable isotope analyses: deuterium, $\delta^{18}O$ and foliar $\Delta^{13}C$**

Branch xylem water was extracted by cryogenic vacuum distillation (~~based on the principals~~ whereby samples are subject to a vacuum and water vapour is frozen using liquid nitrogen, as described in Ingraham and Shadel, 1992; West et al., 2006). Water from each branch sample was extracted for a minimum of 60-75 minutes (West et al., 2006). $\delta^{2}H$ and $\delta^{18}O$ analyses of branch water and groundwater were performed using a Picarro L2120-i Analyser for Isotopic $H_2O$. Five laboratory standards were calibrated against IAEA VSMOW2 – SLAP2 scale (Vienna Standard Mean Ocean Water 2, VSMOW2: $\delta^{18}O = 0$ ‰ and $\delta^{2}H = 0$ ‰; Standard Light Antarctic Precipitation 2, SLAP2: $\delta^{18}O = -55.5$ ‰ and $\delta^{2}H = -427.5$ ‰) and Greenland Ice Sheet Precipitation (GISP, $\delta^{18}O = -24.8$ ‰ and $\delta^{2}H = -189.5$ ‰) as quality control references (IAEA, 2009). The standard

deviation of the residuals between the VSMOW2 – SLAP2 value of the internal standards and the calculated values based on best linear fits was *ca* 0.2 ‰ for $\delta^{18}O$ and *ca* 1.0 ‰ for $\delta^2H$.

**2.4 Carbon isotope ratios of leaves**

Leaf samples stored in paper bags were completely dried in an oven at 60 ℃ for five days. After drying, each leaf sample was finely ground to powder with a Retsch MM300 bead grinding mill (Verder Group, Netherlands) until homogeneous. Between one and two milligrams of ground material was sub-sampled in 3.5 mm X 5 mm tin capsules for analysis of the stable carbon isotope ratio ($\delta^{13}C$), generating three representative independent values per tree. All $\delta^{13}C$ analyses were performed in a Picarro G2121-i Analyser (Picarro, Santa Clara, CA, USA) for isotopic $CO_2$. Atropine and acetanilide were used as laboratory standard references. Results were normalised with the international standards sucrose (IAEA-CH-6, $\delta^{13}C_{VPDB}$ = -10.45 ‰), cellulose (IAEA-CH-3, $\delta^{13}C_{VPDB}$ = -24.72 ‰) and graphite (USGS24, $\delta^{13}C_{VPDB}$ = -16.05 ‰). Standard deviation of the residuals between IAEA standards and calculated values of $\delta^{13}C$ based on best linear fit was *ca* 0.5 ‰.

**2.5 Calculation of WUE$_i$ from $\delta^{13}C$ and hence $\Delta^{13}C$**

WUE$_i$ was determined from $^{13}$C discrimination ($\Delta^{13}C_7$), which is calculated from bulk-leaf carbon isotope ratio ($\delta^{13}C_7$), using the following equations (Werner et al., 2012):

$$\Delta^{13}C \ (‰) = \frac{R_a - R_p}{R_p} = \frac{\delta^{13}C_a - \delta^{13}C_p}{1 + \frac{\delta^{13}C_p}{1000}} \qquad (1)$$

$$WUE_i = \frac{c_a(b - \Delta^{13}C)}{1.6(b - a)} \qquad (2)$$

**2.6 Leaf vein density**

A small sub-section (approximately 1 cm$^2$) of all leaves sampled for $^{13}$C were used for LVD analysis, providing three leaf sub-sections per tree, nine samples per species. Due to the small size of *A. aneura* phyllodes, several phyllodes were combined and ground for $^{13}$C analysis and one whole phyllode was used for LVD analysis. Each leaf section was cleared and stained following the approach described in Gardner (1975). A 5% (w/v) NaOH solution was used as the principal clearing agent. Leaf-sections were immersed in the NaOH solution and placed in an oven at 40 ℃ overnight (Gardner, 1975).

Phyllodes of *Acacia* proved difficult to clear effectively and were kept in the oven longer than overnight to aid clearing. Once cleared, the partially translucent leaf sections were stained with a 1% (w/v) safranin solution. Most leaf sections were stained for up to three minutes and then soaked with a 95% (w/v) ethanol solution until the vein network was sufficiently stained and the majority of colour was removed from the lamina. After staining, the cuticle was removed to aid in identifying the vein network. Following cuticle removal, leaf sub-sections were photographed using a Nikon microscope (model: SMZ800) at 40 x magnification. Finally, minor veins were traced by hand and LVD was calculated as total vein length per unit area (mm/mm$^2$) using ImageJ version 1.48 (National Institutes of Health, USA).

**2.7 Data and statistical analysis**

Species-mean values (n = 9) for the dominant overstorey species at each location were calculated for $\delta^2H$, $\delta^{18}O$, $\delta^{13}C$ and LVD. Relationships between $\Delta^{13}C$ and $WUE_i$ with DTGW were tested using regression analysis after testing for non-normality (Shapiro-Wilk test, $\alpha = 0.05$) and homogeneity of variances (Bartlett test). A Tukey's *post-hoc* test for multiple comparisons across sites was used to test for significance of variation as a function of DTGW. Breakpoints in functions with DTGW were determined using segmented regression analyses whereby the best fitting function is obtained by maximising the statistical coefficient of explanation. The least squares method was applied to each of the two segments while minimising the sum of squares of the differences between observed and calculated values of the dependent variables. Next, one-way ANOVA was applied to determine significance of regressions and breakpoint estimates within a given season. We fully acknowledge that the number of plots with shallow DTGW was sub-optimal, but constraints arising from species distribution across the TI-Tree precluded additional sampling.

**3 Results**

**3.1 Meteorological conditions during the study period**

Mean daily temperature and mean daily vapour pressure deficit (VPD) were largest in summer (December – February) and smallest in winter (June – August) (Figure 1). Daily sums of rainfall showed that the DTW 8.8 m and 49.4 m sites received 265 mm and 228 mm rainfall, respectively, between the September 2013 and April 2014 sampling dates, representing more than 73% and 60% of the total respective rainfall received from January 2013 to June 2014 at these sites.

**3.2 Variation in source-water uptake**

Xylem water isotope ratios for *A. aneura* were widely divergent from the bore water stable isotope ratios (Fig. 2a) in both wet and dry-seasons, indicative of lack of access to groundwater. In contrast, $\delta^2H$ and $\delta^{18}O$ of xylem water in *E. camaldulensis* and *C. opaca* were predominantly (with two exceptions) tightly clustered around $\delta^2H$ and $\delta^{18}O$ of groundwater in the bores located within 50 m of the trees (Fig. 2b). There was little variation in xylem water composition

between the end of the dry and end of the wet-seasons for either species, reflecting the consistent use of groundwater by these two species.

$\Delta^{13}C$ of *A.* ~~aneura~~*aneura, sampled in the Corymbia savanna and Mulga plots* was not significantly correlated with DTGW, in either season or across all values of DTGW (ANOVA F = 1.78; p > 0.05; Fig. 3a, b). As a consequence of these patterns in $\Delta^{13}C$, $WUE_i$ did not vary significantly for *A. aneura* across sites differing in DTGW (Fig. 4a) despite the large variability in $WUE_i$ (ranging 62 – 92 µmol mol$^{-1}$) across sites and seasons. By contrast, foliar $\Delta^{13}C$ of *E. camaldulensis* and *C. opaca* declined significantly with increasing DTGW in both seasons at the sites where DTGW was relatively shallow (DTGW < *ca* 12 m). Segmented regression analysis shown in Figure 3 (c) and (d) yielded breakpoints at 11.17 ± 0.54 m in September (ANOVA F = 11.548; df = 2, 38; P < 0.01) and 9.81 ± 0.4 m in April (ANOVA F = 14.67; df = 2, 47; P < 0.01), which represent the seasonal maximum depths from which groundwater can be extracted by these species. Thus, $WUE_i$ of *E. camaldulensis* and *C. opaca* was significantly smaller at the shallowest site ~~that~~than at sites with DTGW > 13.9 m (Fig. 4b) but did not differ significantly across the deeper DTGW range (13.9 – 49.5 m; Fig. 4b).

~~As distance from Allungra creek increased, foliar~~ $\Delta^{13}C$ for both *E. camaldulensis* and *C. opaca* declined significantly (Fig. 5a), with concomitant increases in $WUE_i$ (Fig. 5b). ~~Fit of the significant regressions in Figure 5(a) improved when applied to either species individually (data not shown; $R^2$ > 0.98).~~ There was no significant relationship for $\Delta^{13}C$ nor $WUE_i$ with distance from the creek for *A. aneura* (data not shown).

### 3.3 Leaf vein density

Leaf vein ~~density~~~~LVD~~density (LVD) did not vary significantly with increasing DTGW for *A. aneura* (Fig. 6a), but a significant increase was observed for DTGW < *ca* 10 m in the two deep-rooted species (Fig. 6b). The break point for *E. camaldulensis* and *C. opaca* (Fig. 6b; 9.36 m ± 0.6 m; ANOVA~~: 3, 59;~~ F = 6.38; P<0.05) agreed well with the two previous estimates (cf Figs. 3 and 6), although again, constraints imposed by species distributions severely limited the number of samplings available at shallow DTGW sites.  As with LVD and DTGW, no relationship was observed between LVD and $\Delta^{13}C$ for *A. aneura* (Fig. 7a), but a significant linear decline in $\Delta^{13}C$ with increasing LVD was observed for *E. camaldulensis* and *C. opaca* (Fig. 7b).

### 4 Discussion

Analyses of stable isotopes of bore water (i.e groundwater) and xylem water across a DTGW gradient established that *A. aneura* adopted an "opportunistic" strategy of water use and was dependent on rainfall stored within the soil profile. This is consistent with previous studies, where Mulga was shown to be very responsive to changes in upper soil moisture content, as expected given their shallow rooting depth and the presence of a shallow (< 1.5 m) hardpan below stands of Mulga (Eamus et al., 2013; Pressland, 1975). Furthermore, very low predawn leaf-water-potentials (~~(~~< -7.2 MPa; Eamus et al., 2016) and

very high sapwood density (0.95 g cm$^{-3}$; Eamus et al., 2016) in *A. aneura* of the Ti Tree and which are strongly correlated with aridity) in *A. aneura* of the Ti Tree, confirms that they rely on soil water without access to groundwater, consistent with the findings of Cleverly et al.,. (2016b). By contrast, analyses of stable isotopes in groundwater and xylem water of *E. camaldulensis* and *C. opaca* established their access to groundwater, as has been inferred previously because of their large rates of transpiration in the dry-season and consistently high (close to zero) predawn water potentials (Howe et al., 2007; O'Grady et al., 2006a; O'Grady et al., 2006b). Importantly, we observed no significant change in xylem isotope composition for these two deep-rooted species between the end of the wet-season and the end of the dry-season, further evidence of year-round access to groundwater at the shallowest DTGW sites.

One specific aim of the present study was to determine whether discrimination against $^{13}$C ($\Delta^{13}$C) and resultant intrinsic water-use efficiency (WUE$_i$) could be used to identify access to groundwater. An increase in foliar $\Delta^{13}$C represents decreased access to water and increasing WUE$_i$ (Leffler and Evans, 1999; Zolfaghar et al., 2014; Zolfaghar et al., 2017). The shallow-rooted *A. aneura* did not show any significant relationship of $\Delta^{13}$C with DTGW during either season. Consequently mean WUE$_i$ showed no significant trend with increasing DTGW, consistent with the conclusion that *A. aneura* only accessed soil water during either season. In contrast, $\Delta^{13}$C of groundwater-dependent vegetation (*E. camaldulensis* and *C. opaca*) declined significantly along the DTGW gradient to a threshold value of DTGW of between 9.4 m (derived from LVD results) and 11.2 m (derived from $\Delta^{13}$C analyses in Sept). Where DTGW was below the threshold (DTGW > *ca* 12m), $\Delta^{13}$C became independent of DTGW. Consequently, WUE$_i$ increased significantly in *E. camaldulensis* and *C. opaca* as DTGW increased from 4.4 m to these thresholds (p < 0.001) but did not vary with further increases in DTGW. We therefore conclude that foliar $\Delta^{13}$C (or WUE$_i$) can be used as an indicator of groundwater access by vegetation. $\Delta^{13}$C is less expensive and easier to measure than stable isotope ratios of water ($\delta^2$H and $\delta^{18}$O) in groundwater, soil water and xylem water. Furthermore, canopies are generally more accessible than groundwater. Globally, identification of groundwater-dependent ecosystems has been hindered by the lack of a relatively cheap and easy methodology (Eamus et al., 2015), thus $\Delta^{13}$C shows great promise for identifying groundwater-dependent vegetation and ecosystems.

Whilst acknowledging the sub-optimal distribution of samples within the shallow DTGW range (< 10 m) from which break-points in regressions were calculated (Figs. 3, 6), which arose because of the natural distribution of trees across the basin, we can ask the question: are the speculated depths beyond which groundwater appears to become inaccessible supported by other independent studies of Australian trees? Eamus et al., (2015) present the results of a seven site (seven sites across the range 2.4 – 37.5 m DTGW), 18 trait, 5 species study and identify a breakpoint between 7 – 9 m. Similarly, two recent reviews identify lower limits to root extraction of groundwater of 7.5 m (Benyon et al., 2006) and 8-10 m (O'Grady et al., 2010) while Cook et al., (1998b) established a limit of 8 – 9 m for a Eucalypt savanna. We therefore conclude that our estimates to limit of groundwater accessibility appear reasonable.

Figure 8 shows combined $\Delta^{13}$C from four Australian studies (Miller et al., 2001; Stewart et al., 1995; Taylor, 2008; Rumman~~,~~ et al., 2017), including one continental-scale study of foliar $\Delta^{13}$C (Rumman~~,~~ et al., 2017~~) and~~). A single regression describes the data of all four independent studies. Thus, when rainfall is the sole source of water for vegetation, $\Delta^{13}$C is strongly correlated with annual rainfall. In contrast, the mean $\Delta^{13}$C for *E. camaldulensis* and *C. opaca* ~~obtained in the present study. $\Delta^{13}$C of *E. camaldulensis* and *C. opaca* did~~do not conform to the ~~regressions of the four independent data sets relating $\Delta^{13}$C to rainfall.~~regression (Fig. 8). It appears that *E. camaldulensis* in the Ti Tree "behaves" as though it were receiving approximately 1700 mm of rainfall, despite growing at a semi-arid site (*ca* 320 mm average annual rainfall). This represents the upper limit to groundwater use by this species, assuming a zero contribution from rainfall (which is clearly very unlikely). The upper limit to annual groundwater use for *C. opaca* was similarly estimated to be 837 mm (Fig. 8). If all of the water from rainfall is used by these two species (which is also very unlikely) then the lower limit to groundwater use is the difference between rainfall and the estimates derived from Figure 8, about ~~1400~~1380 mm for *E. camaldulensis* and ~~500~~517 mm for *C. opaca*.

There any independent estimates of annual tree water-use for these two species which provide a valuable comparison to the estimates made above. O'Grady et al.~~,~~. (2009) showed that annual water use by riparian *E. camaldulensis* in the Ti-Tree was approximately 1642.5 $m^3$ $m^{-2}$ sapwood $y^{-1}$. Assuming average tree radius is 20 cm, sapwood depth is 2 cm and average canopy coverage of the ground is 25 $m^2$ per tree, this yields an annual water use of 1568 mm $y^{-1}$, encouragingly close to the estimate (1700 mm) derived from Figure 8. The estimate for annual water use by *C. opaca* from O'Grady et al (2009) is 837 mm, in reasonable agreement with the estimate from the average $\Delta^{13}$C of *C. opaca* and the regression in Figure 8 (*ca* 900 mm). ~~Depth~~ Because depth-to-~~-~~groundwater for *C. opaca* is significantly larger (*ca* 8 -10 m) than that for *E. camaldulensis* (*ca* 2 - 4 m), ~~thus a smaller rate of groundwater uptake for *C. opaca* is expected since total xylem resistance to flow is presumably larger for the deeper DTGW experienced by this species.~~the resistance to water flow imposed by the xylem's path length is larger for *C. opaca* than *E. camaldulensis* and therefore water-use may be expected to be smaller in the former than the latter, as observed.

Using an entirely different methodology from that used here, O'Grady et al.~~,~~. (2006c) estimated annual groundwater use by riparian vegetation on the Daly River in northern Australia to be between 694 mm and 876 mm, while O'Grady and Holland, (2010) showed annual groundwater use to range from 2 mm to > 700 mm in their continent-wide review of Australian vegetation. Therefore our estimates based on $\Delta^{13}$C appear reasonable. We conclude that (a) *E. camaldulensis* and *C. opaca* are accessing groundwater (because annual water use greatly exceeded annual rainfall); (b) $\Delta^{13}$C can be used as an indicator of groundwater use by vegetation; and (c) $\Delta^{13}$C can provide estimates of upper and lower bounds for the rate of ground water use by vegetation.

### 4.1 Patterns of carbon isotope discrimination and intrinsic water-use efficiency along Allungra creek transects

For the two deep-rooted species, a significant decline in $\Delta^{13}C$ (and hence an increase in WUE$_i$) was observed with increasing distance from the creek (Fig. 6a, b). In contrast to results of O'Grady et al.,. (2006c) for a steeply rising topography in the Daly River, this is unlikely to be attributable to increased elevation since the change in elevation was minimal across each entire transect (< 3 m), and even smaller near Allungra Creek where most of the change in $\Delta^{13}C$ was recorded. Therefore the cause of the change in $\Delta^{13}C$ with distance from the creek was most likely to be a function of the frequency with which trees receive flood water and hence the amount of recharge into the soil profile (Ehleringer and Cooper, 1988; Thorburn et al., 1994; Villeneuve et al., 2015; Singer et al., 2014).  We therefore further conclude that foliar $\Delta^{13}C$ can be used as an indicator of access to additional water to that of rainfall, regardless of the source of that additional water (e.g., groundwater, flood recharged soil water storage, irrigation).

### 4.2 Leaf vein density across the depth-to-groundwater gradient

In our study, LVD was independent of DTGW for *A. aneura* but increased with DTGW for *E. camaldulensis* and *C. opaca* to a break point of 9.4 m, above which LVD was independent of DTGW. Increased DTGW reflects a declining availability of water resources (Zolfaghar et al., 2015), especially in arid-zones.  Uhl and Mosbrugger (1999) concluded that water availability is the most important factor determining LVD.  Sack and Scoffoni (2013) also showed LVD to be negatively correlated with mean annual precipitation in a 796 species meta-analysis. Therefore increasing LVD with increasing DTGW is consistent with the positive effect of water supply on LVD, despite similar amounts of rainfall being received along the DTGW gradient.

Both *A aneura* and *C. opaca* in the present study showed LVDs close to the higher end of the global spectrum (Sack and Scoffoni, 2013), consistent with larger LVDs observed in "semi-desert" species. Higher LVDs allow for a more even spatial distribution of water across the phyllode or lamina during water-stress, which contributes to a greater consistency of mesophyll hydration in species of arid and semi-arid regions (Sommerville et al., 2012). In turn, this allows continued photosynthetic carbon assimilation during water-stress (Sommerville et al., 2010). Presumably, a large LVD also decreases the resistance to water flow from minor veins to mesophyll cells, which is likely to be beneficial for leaf hydration as water availability declines while also facilitating rapid rehydration following rain in these arid-zone species.   Large LVDs for *A. aneura* of semi-arid regions in Australia have been associated with rapid up-regulation of phyllode function with the return of precipitation following drought (Sommerville et al., 2010), and such rapid up-regulation is crucial for vegetation in regions with unpredictable and pulsed rainfall like the Ti Tree (Byrne et al., 2008; Grigg et al., 2010).

LVD was negatively correlated with bulk-leaf $\Delta^{13}C$ (and thus positively correlated with WUE$_i$) in *E. camaldulensis* and *C. opaca*. What mechanism can explain the significant relationship observed between a structural leaf-trait (LVD) and a

functional trait (WUE$_i$)?  The stomatal optimization model (Medlyn et al., 2011) is based on the fact that transpiration (E) and CO$_2$ assimilation (A) are linked via stomatal function. In order to gain carbon most economically while minimising water loss (i.e., optimization of the ratio A/E), stomata should function such that the marginal water cost of carbon assimilation ($\frac{\partial A}{\partial E}$) remains constant (Cowan and Farquhar, 1977; Farquhar and Sharkey, 1982). This aspect of stomatal control

couples the structural traits involved with water-flow with traits associated with primary production (Brodribb and Holbrook, 2007) and explains observed correlations between K$_{leaf}$ and A$_{max}$ in a number of studies (Brodribb et al., 2007; Brodribb et al., 2010; Brodribb et al., 2005; Brodribb and Jordan, 2008; Sack and Holbrook, 2006; Sack and Scoffoni, 2013). The length of hydraulic pathway is directly proportional to K$_{leaf}$ (Brodribb et al., 2007) and the A/g$_s$ ratio determines foliar Δ$^{13}$C (and WUE$_i$) signatures in leaves. Thus, the constraint on K$_{leaf}$ by LVD affects the coordination between the processes of A and E

and thereby might explain significant relationships between structural (LVD) and functional traits (WUE$_i$). For the two deep-rooted species, having access to groundwater resulted in convergence to a common solution for optimizing water supply through veins with respect to E (Brodribb and Holbrook, 2007).

Several robust and testable predictions arise from the conclusion that *E. camaldulensis* and *C. opaca* are functioning at a semi-arid site as though they have access to *ca* 1700 or 900 mm rainfall, respectively. Species growing in high rainfall zones

possess a suite of traits, including low density sapwood, large diameter xylem vessels, small resistance to vessel implosion, large SLA and a large maximum stomatal conductance, compared to species growing in arid regions (O'Grady et al., 2006b; O'Grady et al., 2009; Wright et al., 2004). Therefore, these attributes should be present in *E. camaldulensis* and *C. opaca* if they are functioning as though they are growing in a mesic environment. We have previously established (Eamus et al., 2016; Santini et al., 2016) that these predictions are confirmed by field data and therefore conclude that analyses of foliar

Δ$^{13}$C has global application to the preservation and understanding of GDEs.

## 4.3 Conclusions

We posed five questions regarding depth-to-groundwater (DTGW), foliar discrimination against $^{13}$C (Δ$^{13}$C) and leaf vein density (LVD) as underpinning the rationale for this study. We confirmed that access to shallow groundwater by *E. camaldulensis* and *C. opaca* (DTGW *ca* 0 –11 m) resulted in smaller WUE$_i$ than in *A. aneura*.  We also demonstrated that

LVD correlated with DTGW for the shallower depths (< 10 m) in *E. camaldulensis* and *C. opaca*, but not in *A. aneura*.  We further demonstrated that there was correlation between LVD and Δ$^{13}$C (and hence WUE$_i$) for *E. camaldulensis* and *C. opaca*, but not in *A. aneura*. Similarly, as distance increased from a creek near a flood-out associated with aquifer recharge, foliar Δ$^{13}$C increased and WUE$_i$ decreased for *E. camaldulensis* and *C. opaca,* but not for *A. aneura*. Finally, we conclude that foliar Δ$^{13}$C can be used as an indicator of utilisation of groundwater or stored soil water by vegetation in arid regions,

providing an inexpensive and rapid alternative to the stable isotopes of water that have been used in many previous studies. The observation that the Δ$^{13}$C of the two groundwater-using species was distant from the continental regression of Δ$^{13}$C

against rainfall (Fig. 8) is strong evidence of the value of $\Delta^{13}C$ as an indicator of utilisation of water that is additional to rainfall, and this supplemental water can be derived from either groundwater or soil recharge arising in flood-out zones of creeks.

**5 Author contributions**

All authors contributed to field data sampling. RR undertook all the isotope analyses and statistical analyses and wrote the thesis that formed the basis of this Ms. DE oversaw the design and implementation of the entire project. All authors contributed to writing this Ms and interpretation of the data.

**6 Acknowledgements**

The authors would like to acknowledge the financial support of the Australian Research Council for a Discovery grant awarded to DE.

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

Figures

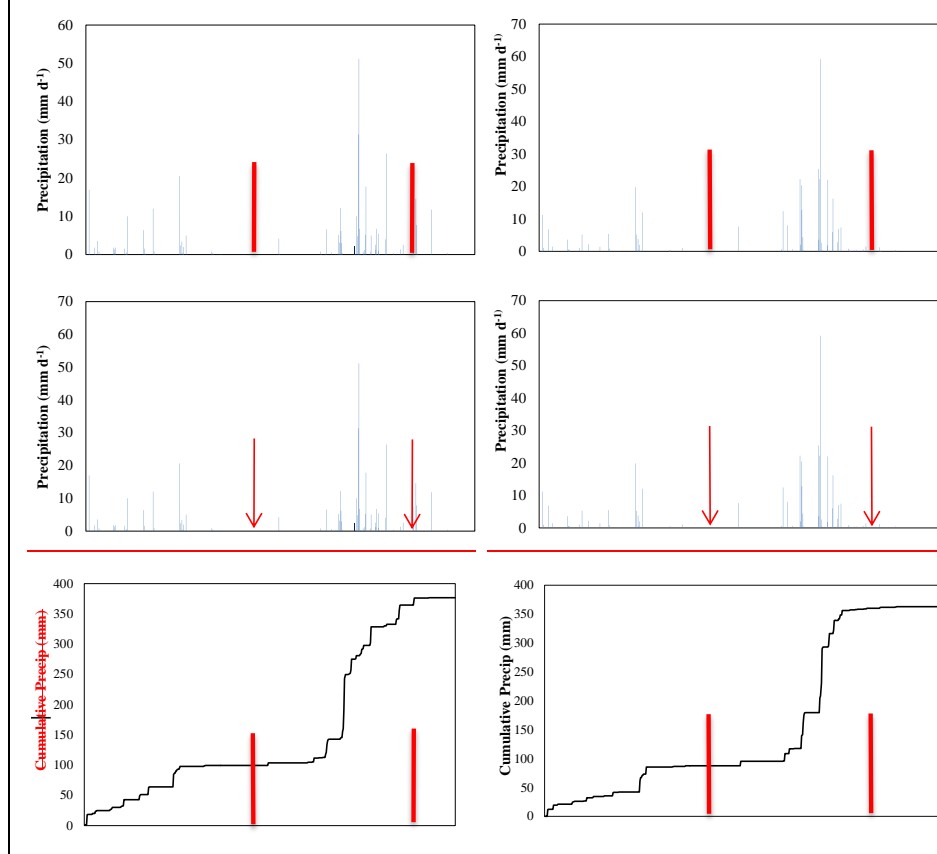

Formatted Table

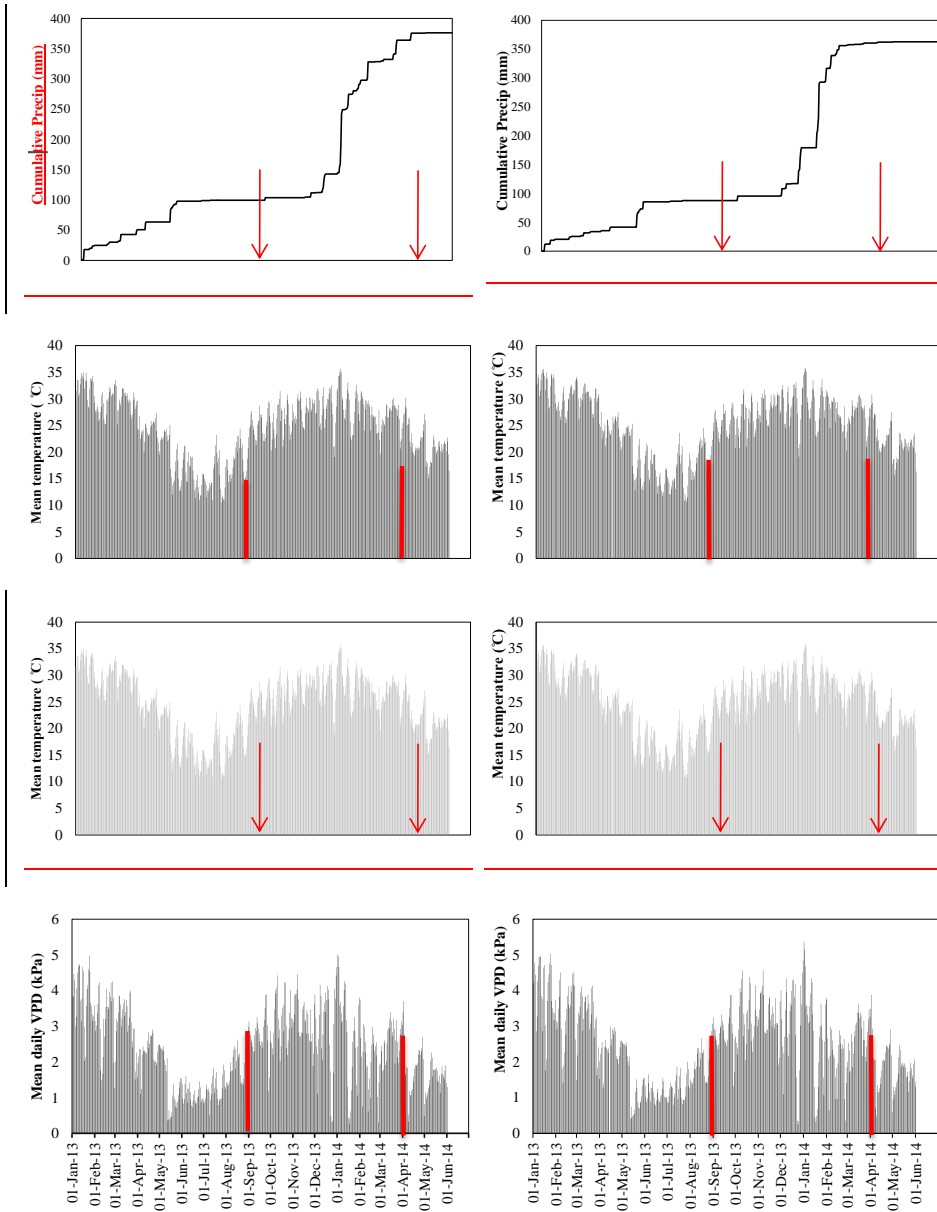

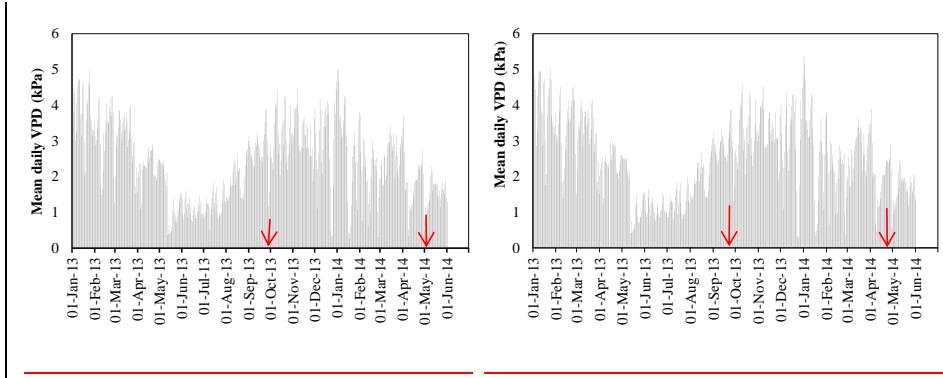

Figure 1:    Mean daily meteorological conditions of daily precipitation, cumulative precipitation, mean air temperature and vapour pressure deficit Jan 2013 - Jun 2014. Red lines show sampling periods in September 2013 (late dry-season) and April 2014 (late wet-season). Left panels show data from the western EC tower (DTGW 49.4 m), and right panels show data for the eastern EC tower (DTGW 8.8 m).

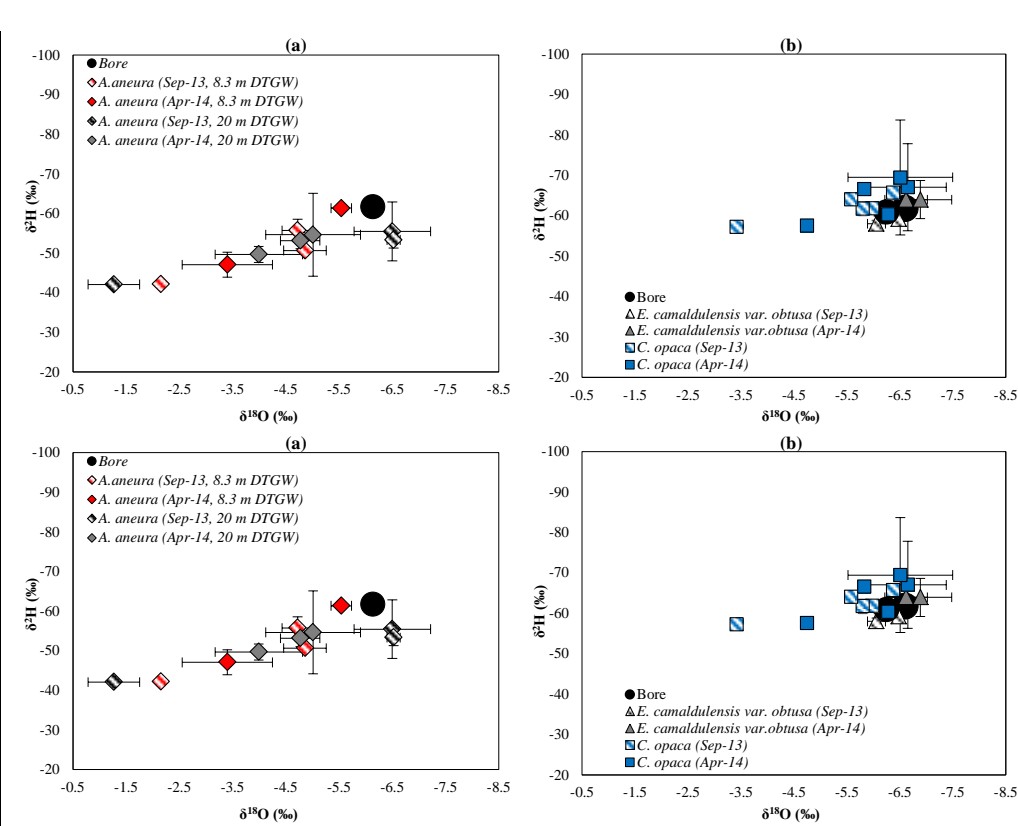

Figure 2: Comparison of xylem and bore water $\delta^2$H-$\delta^{18}$O plots for *Acacia aneura* sampled from 8.3 m and 20 m DTGW (a) and *Eucalyptus camaldulensis* and *Corymbia opaca* sampled from 4.4 m, 8.8 m and 13.9 m DTGW (b). Error bars represent ± one standard error.

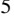

**Sep-2013**                **Apr-2014**

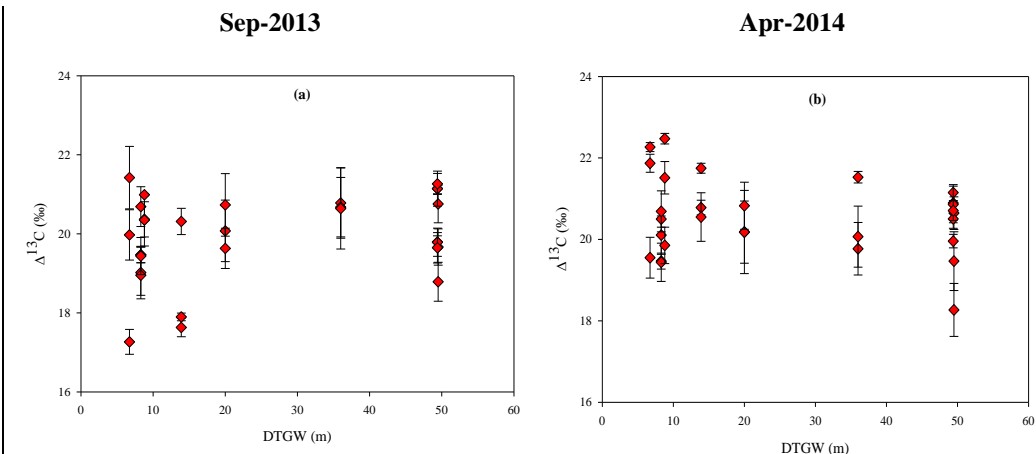

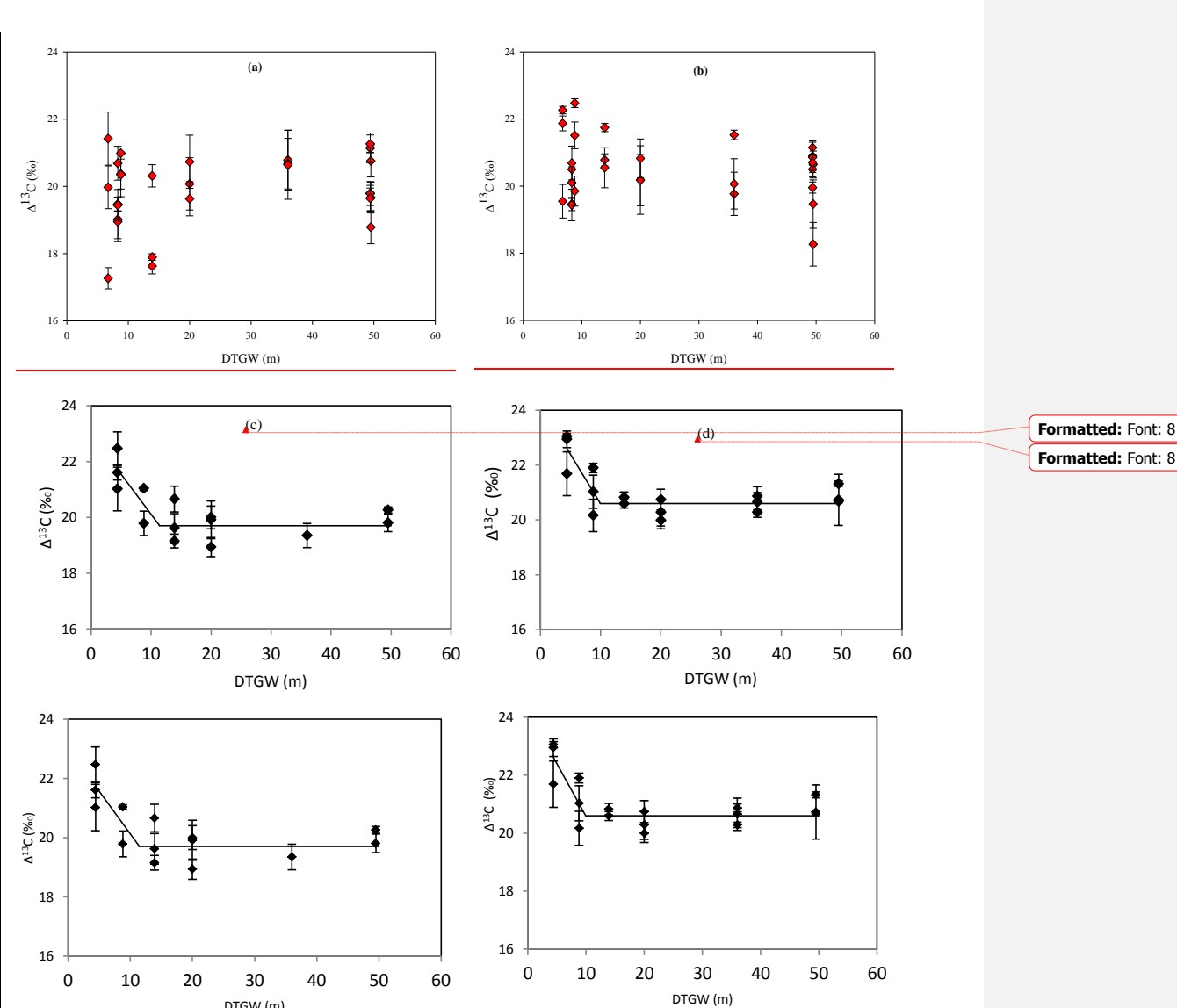

Figure 3:    Carbon isotope discrimination in leaf dry matter ($\Delta^{13}C$) plotted as a function of depth-to-groundwater (DTGW) in the Ti Tree basin. Red points in panel (a) and (b) for *A. aneura* and

black points in panel (c) and (d) *C. opaca* and *E. camaldulensis*. Left and right panels show Sep-2013 and Apr-2014 sampling respectively. Lines in panels (c) and (d) are from segmented regression. Error bars represent ± one standard error.

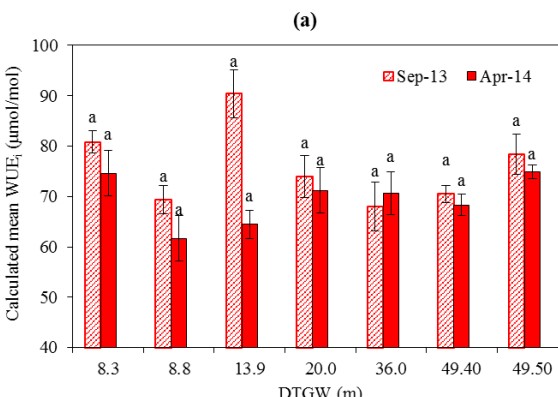

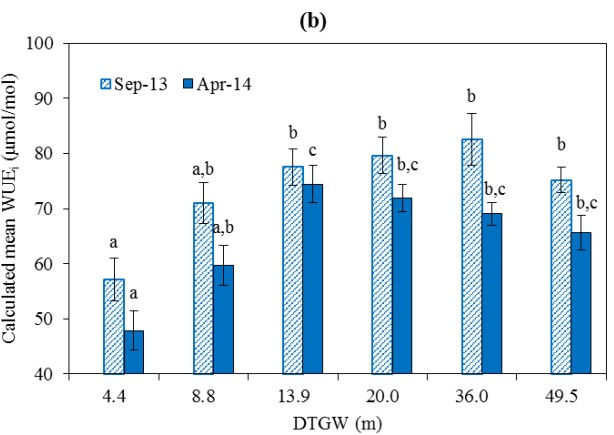

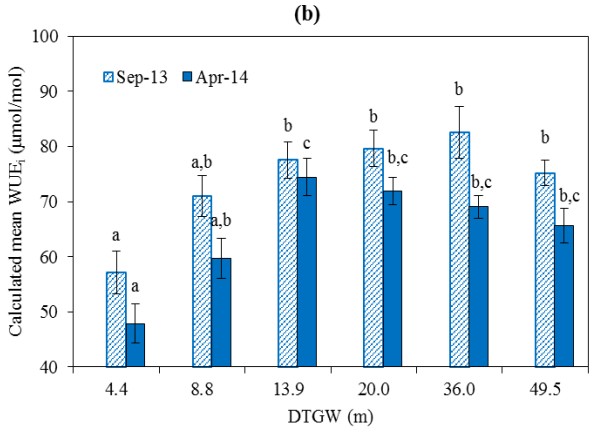

Figure 4:    Leaf intrinsic water-use-efficiency ($WUE_i$) calculated from $\Delta^{13}C$ in shallow-rooted *A. aneura* (a) and deep-rooted *E. camaldulensis* and *C. opaca* (b) across study sites for Sep-2013 (patterned column) and Apr-2014 (filled column). Bars within a season with the same letter are not significantly different across the ~~DTGW~~depth-to-groundwater gradient (Tukey HSD, p < 0.05). Error bars represent ± one standard error.

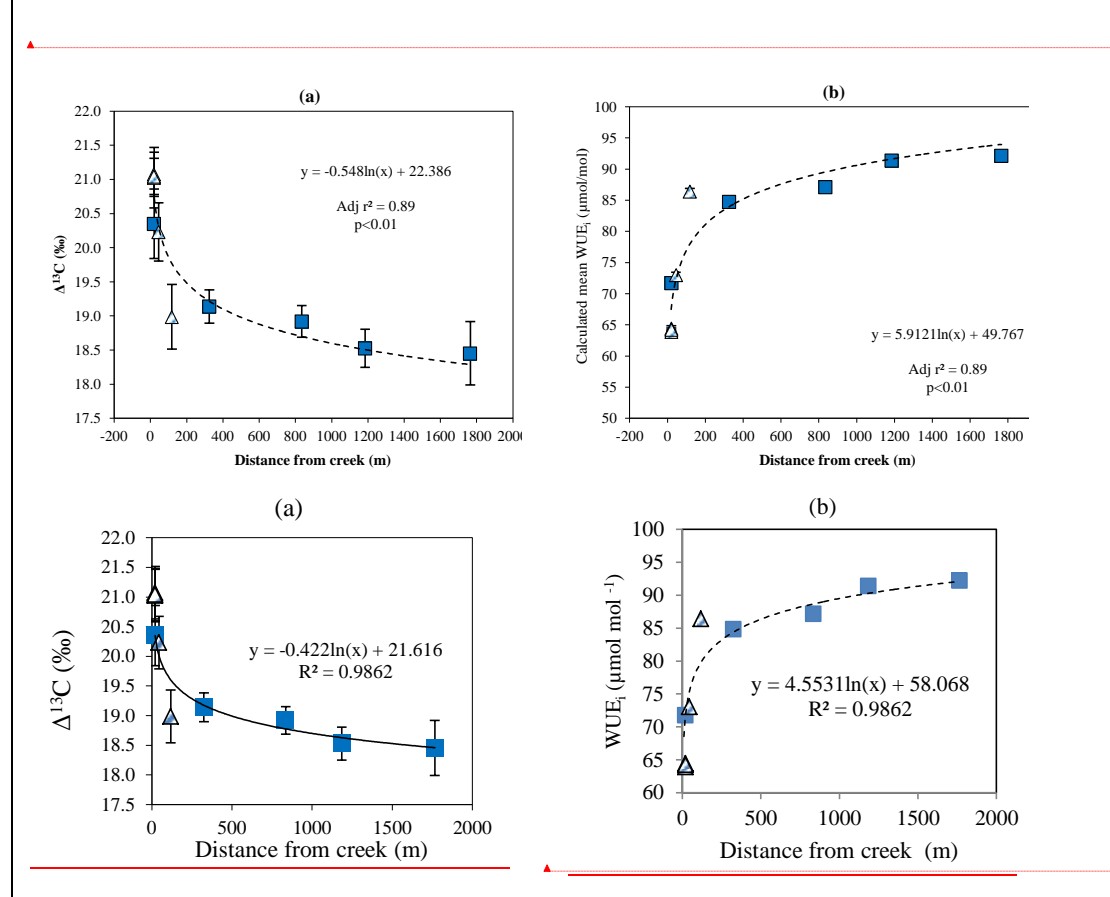

Figure 5:  $\Delta^{13}C$ (a) and $WUE_i$ (b) of deep-rooted species sampled across Ti Tree basin plotted as functions of distance from Allungra Creek bed. Striped triangles represent *E. camaldulensis* and blue squares represent *C. opaca*. Error bars represent ± one standard error. The regression is fitted only to the *E. camuldulensis* data. Note that the largest value of $\Delta^{13}C$ and the lowest value of WUE for *C. opaca* are three overlapping samples.

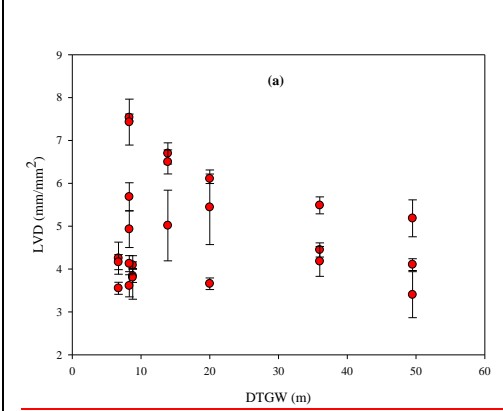

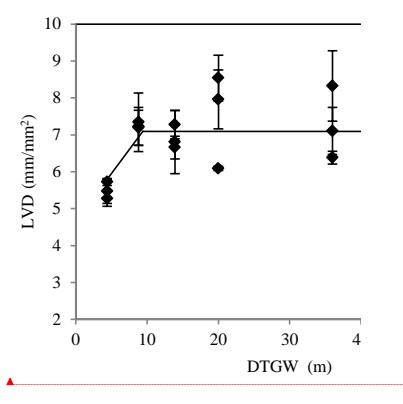

**Formatted Table**

**Deleted Cells**

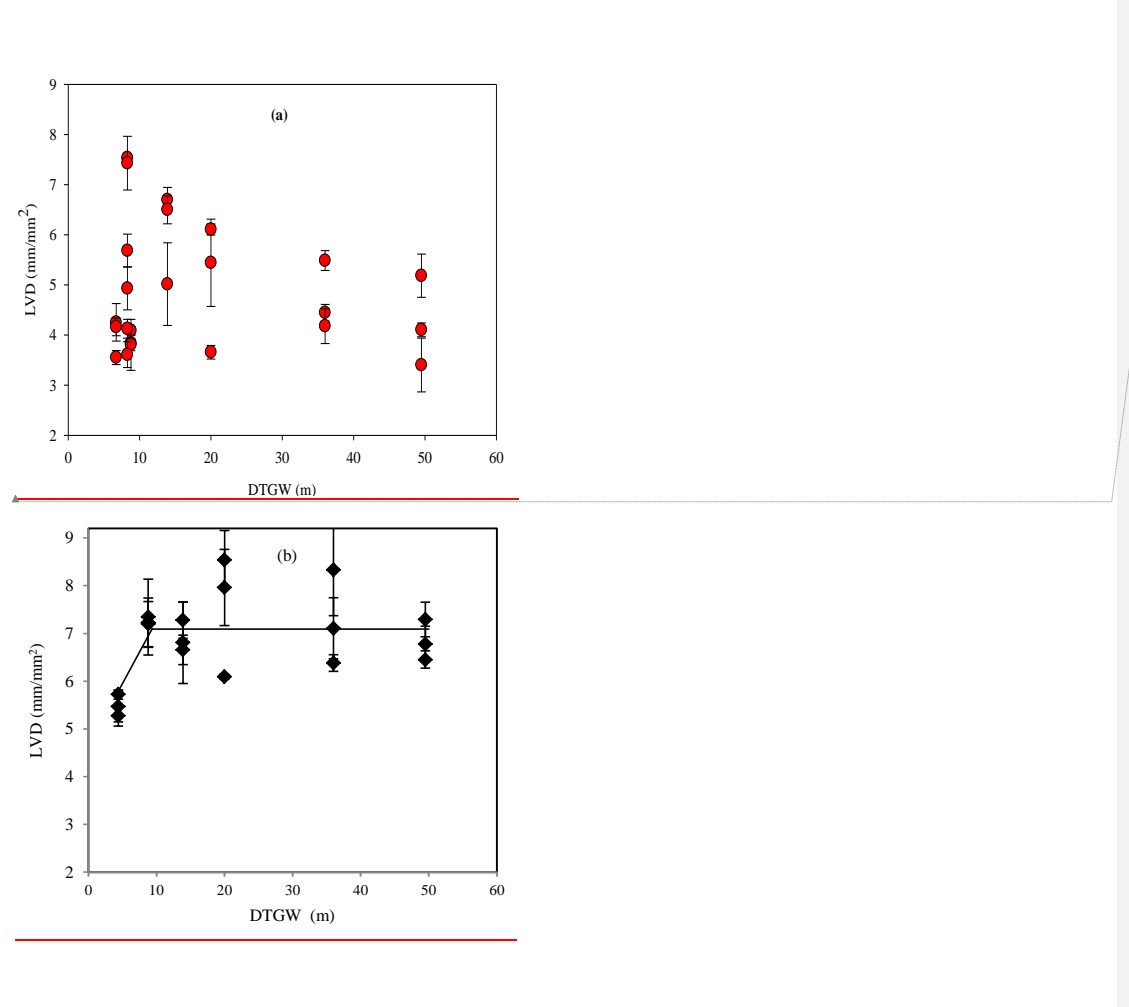

Figure 6:     Leaf vein density (LVD) of *A. aneura* (panel a) or *E. camaldulensis* and *C. opaca* (panel b) as a function of depth-to-groundwater.  Each symbol represents mean LVD calculated from three individual leaves. Error bars represent ± one standard error. A statistically significant correlation derived from segmented linear regression of leaf vein density, for *E. camaldulensis* and *C. opaca*

with depth-to-groundwater (DTGW) is shown in panel (b). The $r^2$ and standard deviation slope of the regression below the break point in (b) are 0.976 a d 0.0031 respectively.

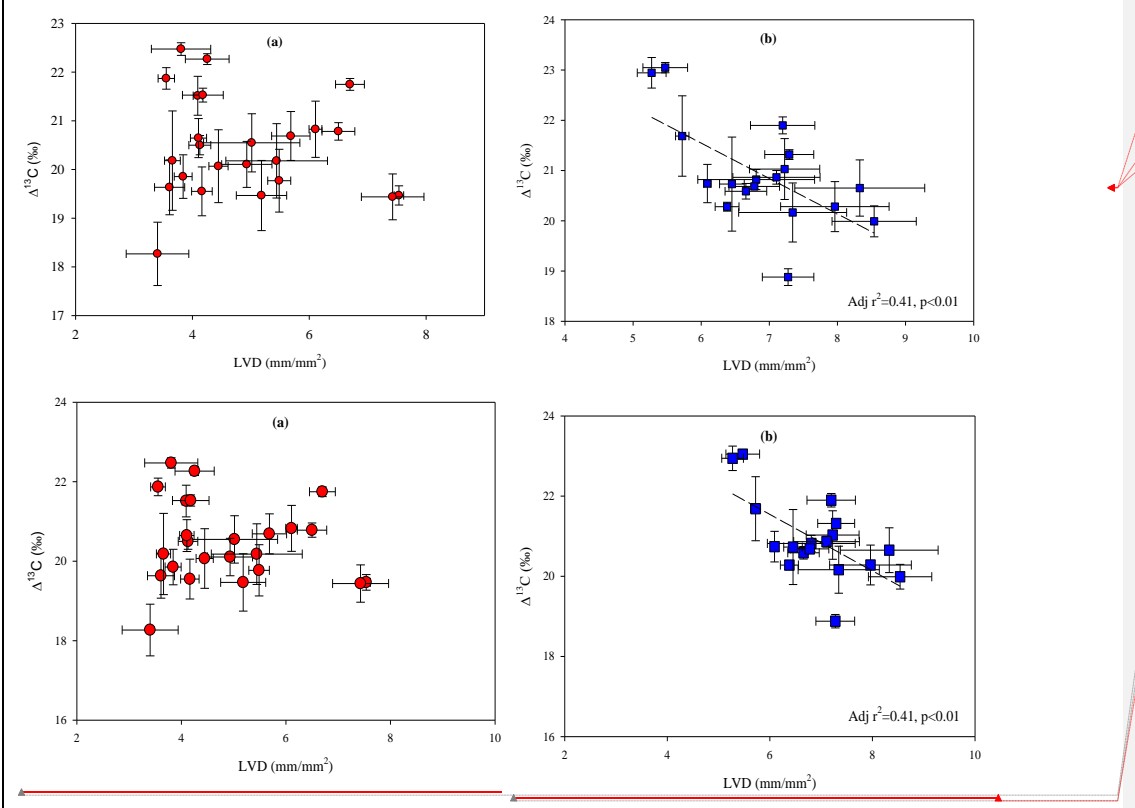

Figure 7:    Relationships of leaf vein density of (a) *A. aneura* (a) and (b) *E. camaldulensis* and *C. opaca* (b) with bulk-leaf $\Delta^{13}C$. Each symbol represents mean LVD and $\Delta^{13}C$ calculated from, with both variables measured on the same individual leavesleaf. Error bars represent ± one standard error. A statistically significant correlation of LVD and $\Delta^{13}C$ of *E. camaldulensis* and *C. opaca* is plotted with a dashed line.

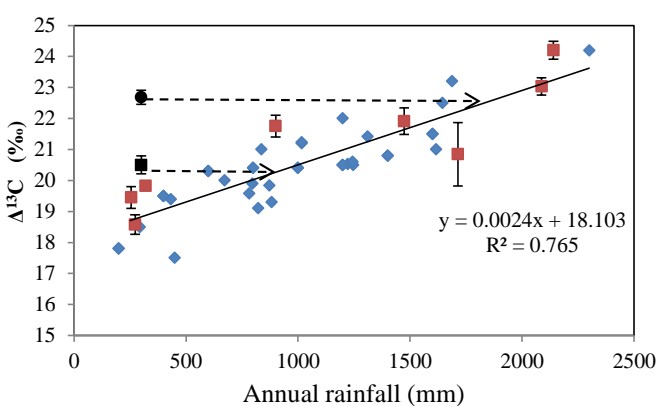

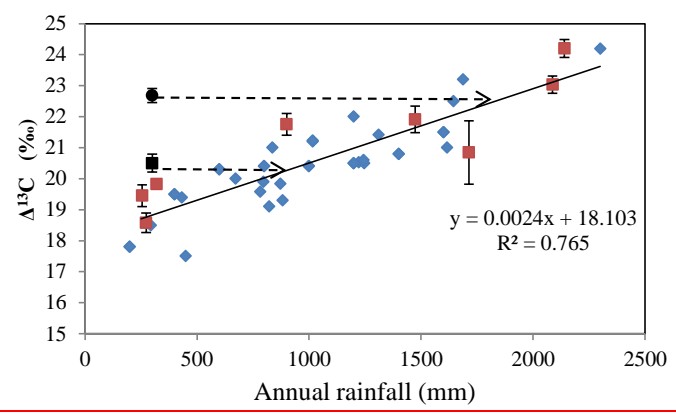

Figure 8: Relationships of discrimination against carbon-13 ($\Delta^{13}$C) with annual rainfall observed in different studies across Australia. The diamonds represent observations made in eastern Australia (Stewart et al., 1995), northern Australia (Miller et al., 2001) and sites in New South Wales (Taylor, 2008). The red squares are data from a continental-scale assessment of foliar $\Delta^{13}$C (Rumman et al., 2017). The black circle is the mean $\Delta^{13}$C of *E. camaldulensis* and the black square is the mean $\Delta^{13}$C *C. opaca*, both of which were measured in this study. The 95 % CI for the mean $\Delta^{13}$C is + 0.403 and the s.e of the slope is 0.000231. The black arrows indicate the rainfall that would be required to account for the $\Delta^{13}$C for *E. camaldulensis* and *C. opaca* if these two species relied only upon rainfall.