# Peer review of "Speculations on the application of foliar $^{13}\text{C}$ discrimination to reveal groundwater dependency of vegetation, provide estimates of root depth and rates of groundwater use"

_Hydrology and Earth System Sciences, 2017_

## Short Comment (SC1) · 4 Dec 2017

This is a fascinating study looking at the relationship between leaf traits and depth to groundwater. It is a valuable contribution to the literature because the authors have shown that leaf isotopes and venation patterns can be used as indicators of groundwater use. With ongoing extraction of groundwater across the world, simple methods for monitoring vegetation water use patterns in groundwater dependant ecosystems are very useful. The insights are also ecologically informative and the paper is well-written. I have one question for the authors. The relationships between DTGW and delta 13 C

[Figure]

(Fig. 3c,d) and LVD (Fig. 6b) are shown as broken stick relationships but the relationship between delta 13 C and LVD (Fig. 7b) is shown as a linear regression. How do the authors account for the difference in curve shapes and is a linear regression the best model for Fig. 7b? Minor comments A description of components of equations 1 and 2 would be helpful. I would prefer to see arrows instead of red lines on Fig. 1 as the red bars cover some of the data. In Fig. 2, different points are shown as Sept or April but in the text, they are described as wet and dry. I suggest explaining wet and dry seasons in the caption of Fig. 2. The same applies for Fig. 3.

---

## Author Comment (AC1) · 18 Dec 2017

We would like to than Dr Macinnis-Ng for taking the time to comment on our speculative paper. We thank her for her kind comments about the "fascinating study" and "valuable contribution".

We believe that the lack of a broken stick relationship in Fig 7 arises because in Fig 3, there is seasonal variation in 13C and in Fig 6 there is no seasonal variation in LVD (nor depth-to-groundwater in either figure). Consequently when we plot the annual

mean 13C against the corresponding LVD the variation in 13C results in an inability to identify a significant broken stick relationship in Fig 7. A linear relationship in Fig 7 was the best significant fit to the data.

We thank Dr Macinnis-Ng for the suggestion to explain the meaning of the symbols in Eq 1, 2 and we shall make the changes in the final version of the Ms. We shall also explain that wet and dry seasons correspond to April and Sept sampling periods, in the revised Ms. We don't believe the red bars preclude the value of Fig 1 as daily values of VPD or temp are not of material value to the Ms, only the seasonal trends.

Once again we thank Dr Macinnis-Ng for her helpful and supportive comments.

---

## Referee Comment (RC1) · Anonymous Referee #1 · 8 Feb 2018

This is a rigorous study examining the feasibility of using $\Delta$13C as an indicator of groundwater use by vegetation. The paper is well written and addresses the important issue of determining vegetative water sources. The potential to use the measurement of $\partial$13C in place of the more time consuming $\partial$2H and $\partial$18O analyses is promising and a very useful contribution to the field.

Overall, the paper is well organized and interesting to read. One thing I might suggest is a diagram or table depicting the different sampling plots/transects and how many/what kind of samples were taken in each. This is not completely necessary, but with such a

wide variety of variables being examined, it might be helpful.

Minor comments:

Figure 1: Change red lines to arrows (or something else). Right now they are covering data and look misleadingly like they are depicting actual data. Also, the y-axes in the top two panels do not match, but are consistent between all other pairs of panels.

Page 11, line 9: Error "shallowest site that at sites"

Figure 3: Make fonts and panel sizes consistent

Figure 5: X-axis label is cut off in panel (a)

Figure 5: It seems odd to combine both species together for the regressions when there are only data from E. camaldulensis for a small fraction of the total distance examined. It would be better just to include the regression for C. opaca while still keeping the E. camaldulensis data points in the figure for reference.

3.3 Leaf vein density: Error in first line "Leaf vein densityLVD"

Figure 6: Make font size in (a) match font size in (b)

Figure 7: Error in figure legend "from same individual leaves"

---

## Referee Comment (RC2) · Anonymous Referee #2 · 8 Mar 2018

The study presented here discusses the potential of delta 13C in leaf tissue as a surrogate for access to groundwater. Additionally, if the depth to groundwater is known information about rooting depth can be inferred. Furthermore, it seems that plant traits – here leaf vein density – allow to retrieve information about water availabilities and access during plant /leaf development. The paper is well written and organized. But, the authors have to check rigorous for typos, comma placement (especially when references are cited!). This study is of high importance for the scientific community. Especially it shows potential use of a simpler method to infer water resource use in waterlimited environments instead of using several isotopes and a relatively high number of samples from several soil depths and plant tissues. I would suggest to include a map in which the Ti Tree Basin and the sites are shown (maybe one showing Australia and inserted / zoomed in to Ti Tree Basin region showing (e.g. add. also showing groundwater reservoir?)). I think, the map would be helpful for non-Australian.

Specific and technical comments:

p.1 line 23: add 'basin' at end of sentence. p.1 line 28: change 'attributes' to 'traits' p.3 line 9: change to 'deep-rooted' p.3 lines 30 f.: Change 'One' to 'Another'. And, Please introduce DTGW – right now it is introduced the following page. p.3 lines 26 ff.: What are the distances (min., max., average) of the meteo station to your sampling sites? p.5 line 7ff.: How deep is the water table situated in the eastern part? p.5 line 21: Is it four sampling plots per site? p.6 line 4: Are the distances stand for? This is not clear to me. p.6 line 9: "2.2 Leaf and water sampling protocols and meteorology" p.6 line 13: How did you collect the bore water samples? Could you briefly describe the collection process? Introduce once wet and dry season (the months) and then stick to these categories. It is easier to follow. p.6 lines 21 ff.: What apparatus did you use for the distillation? Maybe it is useful to describe briefly the extraction process. p.7 line 3: maybe add 'powder' at end of the sentence p.7 line 5: What do you mean with "generating three representative independent values per tree"? You had three leaves samples per tree and measured these independently? p.7 line 12ff.: Please give more information for the formulae? Maybe some reader need info about the difference between the two different notations of 13 C. p.8 line 10: Mention here that you used a (segmented) linear regression. How was the maximizing of $R^2$ done? Results: Maybe you could also provide some values (e.g. for vpd, isotopic signals) in the text beside the figures. p.8 line 18: DTGW ! p.9 line 11 ff.: Does the DTGW decreases with the distance to the creek? Maybe there is a strong relationship between these parameters? p.9 line 12-13: This would be a species-specific relationship. Delete 'either'. p.9 line 16: Leaf vein density (LVD) p.9 line 18: correct the ANOVA value, please. p.9 line

27f: Could you give values for water potentials and sapwood density? p.10 line 6f: Another possibility is that not the water access but the biochemistry is limiting (Rubisco limitation). p.10 line 8ff.: Please mention the Figures in the text where this information can be found would be helpful. p.10 line 15: ...if the species-specific signatures (or 'behavior') under well-watered (or groundwater access) and reduced water availability are known. I would think. p.10 line 29: What species was studied in Rumman (2017)? p.11. line 1-11f.: For me it was not easy to follow during this paragraph. Would it be possible to rephrase this paragraph. I got the point – but it took a bit. Especially this information in here is very important and interesting!

Figures

As I mentioned before, it could be helpful to readers to have a map where Ti Tree basin is located. Figure 1: I would recommend to include the precipitation data (mm d-1) within the cumulative precipitation figures. Eastern and western tower data should have the same scaling (not only for easier comparison). Precipitation (mm d-1) could be shown on right hand side (y-axis). All these panels could be moved closer (no gap in between), esp. for eastern and western tower individually. This saves also space. The use of ticks on the x- and y-axis could also make it easier to 'follow'. Figure 2: Maybe you can change the coordination. Right upper corner would be 0/0 – because x- and y-axis show both negative values. I would also start at 0/0. Figure 3: Symbol and Font sizes should be larger and uniform. Upper and lower panels could be closer (no gap). Maybe change the months to wet and dry season. In the caption insert 'depth-to-groundwater' before 'DTGW'. Figure 4: These panels could also be closer (no gap). I would prefer the DTGW data as continuous and NOT as categorical (same distances) between bars! Maybe the use of scatter plot instead of bar plot would also be more appropriate. Maybe change the months to wet and dry season. In the caption insert 'depth-to-groundwater' before 'DTGW'. Figure 5: Is there a relationship between distance from creek and DTGW? Did you test this? Figure 6: Symbol and Font sizes should be larger and uniform. In the caption insert 'DTGW' after 'depthto-groundwater'. (b) or panel b is missing in panel b. LVD at 9 m DTGW and 12 m DTGE are not significantly different and with increasing DTGW depth LVD increases till 20 m DTGW. What is the R2 and p-value for this linear regression. I am skeptical about this segmented regression on LVD with changing DTGW. Figure 7: Would it be possible to use different colors or symbols for the two different tree species in panel (b)? This would give the reader more information for the two species and their adjustments. Please use similar scaling for (a) and (b). Font and symbol sizes could be a bit larger. Figure 8: It would be great if the authors could show the 95% confidence interval for the linear regression. This would add more information. Please state in the caption what the black arrows stand for.

---

## Author Comment (AC2) · 4 Apr 2018

Response to anonymous referee 2:

We are pleased to see that referee 2 considers the paper to be well written and organized. We also thank the referee for saying that the study to be of high importance. We have followed the referee's advice to check rigorously for typos and the placement of commas. In reply to the referee's comments we note:

a) We have added a map in the Supplementary material and this indicates the site location and the location of the basin as a whole within the Northern Territory of Australia and Australia as a whole.

b) We have added the word "basin" at the end of the sentence.

c) We disagree with changing the word attributes to traits. It isn't the traits per se that is surprising, it is the values those traits have, that is surprising. The value of the trait is an attribute, not a trait. The trait is, for example, SLA. The attribute is "large" or "small values" of this trait.

d) We have inserted the hyphen in deep-rooted.

e) We have modified the wording of line 30 on p3; and DTGW has been defined here too.

f) We have indicated the distance from the Met station to all study sites (line 27).

g) We have indicated GW depth in the eastern region.

h) No, it is four plots, each with a different DTGW, as stated in the methods in several places.

i) We have reworded p6 line 4 to show that the distances refer to the depth-to-groundwater.

j) We have added text to explain how bore water was collected on p6.

k) The method to extract water is amply described by the phrase "cryogenic vacuum distillation" and the citations to the key references of Ingraham and Shadel (1992) and West et al., 2006). We have added some text to refer to the key features, namely imposition of a vacuum and subsequent freezing of water vapour with liquid nitrogen.

l) We have inserted the word "powder" in line 5 of P7.

m) Yes, three leaf samples per tree were measured, independently.

n) We have defined the meaning of the two deltas in equations one and two on p7.

o) Lines 12 – 16 on p8 explain in detail how segmented regressions were undertaken.

p) The abbreviation DTGW has been defined prior to its use on p8.

q) Lines 3 – 12 of p9 does not refer to data sampled along the Allunga Creek. The data presented in this paragraph were from the Corymbia savanna and riparian plots. We have added the words "sampled in the Corymbia savanna and Mulga plots" at the start of the paragraph to make this clearer.

r) We have deleted the word "either" as requested.

s) We have corrected the ANOVA value as requested.

t) We have added the pre-dawn water potential and sapwood density values as suggested, on p9.

u) The possibility of Rubisco limitation for these Acacias is extensively discussed in Nolan et al (2017; Functional Plant Biology 44, 1087 – 1097) and this is not an explanation.

v) Figure numbers are cited as requested.

w) We have amended the citation of Rumman 2017 to Rumman et al 2017 as this is now the published version (Global Change Biology 24, 1186 – 1200). The full species list is available there. It is far too large a list to cite in the HESS Ms.

x) We have rewritten the paragraph mentioned by the referee and hope this is now clearer to the reader. A map of the Ti Tree is now included in the Supplementary material.

z) We have chosen not to add the daily rainfall data to the cumulative sum plot as it makes the plot too "busy". Moving the plots closed together in Fig 1 makes the axes labels become less distinct.

aa) The range of y values in Fig 2 was chosen to optimize the clarity of the plots and

we have chosen to leaves these as they are.

ab) Symbol and font sizes have been reduced in the lower two panels if Fig 3. DTGW has been defined in the legend.

ac) Given the fact that HESS is published on-line, and therefore space is not a critical issue, we have chosen to leave the panels spaced as they are for clarity and ease of reading in Fig 4. We understand the referee would prefer to see DTGW as continuous data, we feel that by the time the reader has got to this figure they will be comfortable with seeing the plots as categorical. We have added "depth-to-groundwater" in the figure legend as requested.

ad) As discussed in section 4.1 of the Discussion, the topographic gradient is very small close to Allungra Creek. Consequently depth-to-groundwater as a function of distance from the Creek shows little variation. What changes with distance from the creek is the amount of recharge of the soil profile following flood events. It was not possible to measure depth-to-groundwater here because there are no bores installed. As we discuss in the Ms, the change in $\Delta 13C$ as a function of distance from the creek is a function of access to additional water arising from bank recharge following flooding, not access to groundwater.

ae) We have inserted "depth-to-groundwater" in the fig legend and increased the font size and data point size in Fig 6. We have added the R2 and the st.dev of the regression below the breakpoint into the figure legend. We think a key point to note is that the segmented regression, when applied to two independent data sets (13$\Delta$C data (Fig. 3) and LVD data (Fig. 6)) yield very similar relationships with very similar breakpoints (as discussed in the MS) and this adds confidence to the speculations made about what these breakpoints represent ecophysiologically.

af) We have redrawn Fig 7 and made the font and data points larger.

ag) It is not possible to plot the 95 % CI on Fig 8 and still retain clarity. Consequently

we have quoted the 95% CI in the legend so that readers can have that value to-hand. We have indicted in the legend what the arrows mean.

Please also note the supplement to this comment: https://www.hydrol-earth-syst-sci-discuss.net/hess-2017-540/hess-2017-540-AC2-supplement.pdf

———————————————————

---

## Author Comment (AC3) · 4 Apr 2018

Response to anonymous referee 1:

We thank the referee for their kind comments about the "useful contribution to the field" and that the paper is well organized and interesting to read.

In response to your comments, we have made the following changes:

a) Figure 1 – replaced the red bar with an arrow; made the y axes in the top two panels

match so that the upper limit is 70 mm d-1 in both panels.

b) Corrected the typo ". ... site that at sites . ..." now reads ". ... sites than at sites . . ."

c) We have made font sizes and panel sizes equal in Fig. 3.

d) We have adjusted the x axis label to prevent the 2000 being partially cut-off.

e) We have applied the regression only to the eucalypt and amended the text in the Ms to reflect this.

f) We have corrected the typographical error in the first line of section 3.3.

g) Font sizes corrected in Fig 6.

---

## Editor Decision (ED1)

[revised manuscript text omitted]

**1 Supplementary Material**

3 Table S1: List of species, their functional types, Δ$^{13}$C and calculated intrinsic WUE$_i$ (plant functional
4 type codes: BlT= broad leaf tree, NlT= needle leaf tree and S=shrub).

| Species | Site | PFT | Δ$^{13}$C (‰) | Calculated WUE$_i$ |
|---|---|---|---|---|
| *Acacia aneura* | AMU | NlT | 19.71±0.21 | 78.67 |
| *Acacia melanoxylon* | WR | BlT | 22.46±0.46 | 49.00 |
| *Acmena graveolens* | CT | BlT | 21.96±0.31 | 54.37 |
| *Alphitonia whitei* | RC | BlT | 24.81±0.58 | 23.63 |
| *Alstonia muelleriana* | RC | BlT | 23.42±0.4 | 38.58 |
| *Alstonia scholaris* | CT | BlT | 22.76±0.36 | 45.75 |
| *Anopterus glandulosus* | WR | S | 25.58±0.44 | 15.34 |
| *Argyrodendron peralatum* | CT | BlT | 24.89±0.26 | 22.75 |
| *Atherosperma moschatum* | WR | BlT | 22.57±0.85 | 47.80 |
| *Cardwellia sublimis* | CT | BlT | 22.46±0.58 | 48.92 |
| *Castanospermum australe* | CT | BlT | 22.04±0.4 | 53.48 |
| *Ceratopetalum succirubrum* | RC | BlT | 26.52±0.13 | 5.21 |
| *Corymbia terminalis* | AMU | BlT | 19.89±0.29 | 76.65 |
| *Cryptocarya mackinnoniana* | CT | BlT | 24.59±0.12 | 26.02 |
| *Daphnandra repandula* | RC | BlT | 23.69±0.57 | 35.69 |
| *Doryphora aromatica* | RC | BlT | 25.41±0.88 | 17.19 |
| *Dysoxylum papuanum* | CT | BlT | 20.83±0.44 | 66.59 |
| *Elaeocarpus grandis* | CT | BlT | 22.34±0.1 | 50.30 |
| *Endiandra leptodendron* | CT | BlT | 26.86±0.6 | 1.51 |
| *Eucalyptus amplifolia* | CP | BlT | 21.02±0.31 | 64.53 |
| *Eucalyptus camaldulensis* | AMU | BlT | 20.28±0.35 | 72.43 |
| *Eucalyptus clelandii* | GWW | BlT | 17.68±0.17 | 100.56 |
| *Eucalyptus dumosa* | CM | BlT | 19.01±0.42 | 86.15 |
| *Eucalyptus fibrosa* | CP | BlT | 23.24±0.32 | 40.57 |

| Species | Site | PFT | $\Delta^{13}C$ (‰) | Calculated WUE$_i$ |
|---|---|---|---|---|
| *Eucalyptus miniata* | LF | BlT | 21.86±0.07 | 55.39 |
| *Eucalyptus moluccana* | CP | BlT | 21.48±0.24 | 59.50 |
| *Eucalyptus obliqua* | WR | BlT | 22.01±0.73 | 53.79 |
| *Eucalyptus salmonophloia* | GWW | BlT | 18.88±0.09 | 87.56 |
| *Eucalyptus salubris* | GWW | BlT | 18.02±0.19 | 96.82 |
| *Eucalyptus socialis* | CM | BlT | 19.94±0.18 | 76.17 |
| *Eucalyptus tereticornis* | CP | BlT | 21.27±0.26 | 61.85 |
| *Eucalyptus tetrodonta* | LF | BlT | 19.82±0.34 | 77.39 |
| *Eucalyptus transcontinentalis* | GWW | BlT | 19.75±0.31 | 78.24 |
| *Eucryphia lucida* | WR | BlT | 22.07±0.06 | 53.16 |
| *Ficus leptoclada* | RC | BlT | 21.53±0.49 | 59.01 |
| *Ficus variegata* | CT | BlT | 21.85±0.29 | 55.60 |
| *Flindersia bourjotiana* | RC | BlT | 23.8±0.43 | 34.54 |
| *Gillbeea adenopetala* | RC | BlT | 24.41±0.26 | 27.94 |
| *Gillbeea whypallana* | CT | BlT | 23.22±0.42 | 40.79 |
| *Leptospermum lanigerum* | WR | BlT | 21.61±0.39 | 58.15 |
| *Litsea leefeana* | RC | BlT | 23.9±0.48 | 33.38 |
| *Melaleuca squarrosa* | WR | BlT | 21.94±0.25 | 54.60 |
| *Myristica globosa* | CT | BlT | 24.04±0.53 | 31.91 |
| *Notelaea ligustrina* | WR | BlT | 21.2±0.51 | 62.52 |
| *Nothofagus cunninghamii* | WR | BlT | 21.06±0.59 | 64.02 |
| *Phyllocladus aspleniifolius* | WR | S | 18.48±0.68 | 91.93 |
| *Pittosporum bicolor* | WR | BlT | 19.99±0.67 | 75.63 |
| *Polyscias elegans* | RC | BlT | 22.96±0.6 | 43.60 |
| *Pomaderris apetala* | WR | S | 23.55±0.44 | 37.21 |
| *Prunus turneriana* | RC | BlT | 25.83±0.74 | 12.59 |
| *Rockinghamia angustifolia* | CT | BlT | 22.69±0.35 | 46.49 |
| *Synima cordierorum* | CT | BlT | 22.8±0.39 | 45.33 |
| *Syzygium johnsonii* | RC | BlT | 25.7±0.07 | 13.99 |

| Species | Site | PFT | $\Delta^{13}C$ (‰) | Calculated WUE$_i$ |
|---|---|---|---|---|
| *Syzygium sayeri* | CT | BlT | 23.01±0.46 | 43.04 |
| *Tasmannia lanceolata* | WR | S | 21.8±0.26 | 56.03 |
| *Xanthophyllum octandrum* | CT | BlT | 22.64±0.6 | 47.08 |

8    **Table S2: List of climate variables used in climate analysis**

| WorldClim Code | Variables |
| --- | --- |
| $BIO_1$ | Mean Annual Temperature |
| $BIO_2$ | Mean Diurnal Range |
| $BIO_3$ | Isothermality |
| $BIO_4$ | Temperature Seasonality |
| $BIO_5$ | Max Temperature of Warmest Month |
| $BIO_6$ | Min Temperature of Coldest Month |
| $BIO_7$ | Temperature Annual Range |
| $BIO_8$ | Mean Temperature of Wettest Quarter |
| $BIO_9$ | Mean Temperature of Driest Quarter |
| $BIO_{10}$ | Mean Temperature of Warmest Quarter |
| $BIO_{11}$ | Mean Temperature of Coldest Quarter |
| $BIO_{12}$ | Mean Annual Precipitation |
| $BIO_{13}$ | Precipitation of Wettest Month |
| $BIO_{14}$ | Precipitation of Driest Month |
| $BIO_{15}$ | Precipitation Seasonality |
| $BIO_{16}$ | Precipitation of Wettest Quarter |
| $BIO_{17}$ | Precipitation of Driest Quarter |
| $BIO_{18}$ | Precipitation of Warmest Quarter |
| $BIO_{19}$ | Precipitation of Coldest Quarter |

**Table S3: Relationships of leaf traits with Mean Annual Precipitation and Moisture Index**

| Traits | Correlated with ln MAP | | | | | | Correlated with MI | | | | | |
|---|---|---|---|---|---|---|---|---|---|---|---|---|
| | Dry-season | | | Wet-season | | | Dry-season | | | Wet-season | | |
| | *Adj r²* | *slope* | *p* | *Adj r²* | *slope* | *p* | *Adj r²* | *slope* | *p* | *Adj r²* | *slope* | *p* |
| $A_{400}$ | 0.224 | + | <0.0001 | NS | | > 0.05 | 0.071 | + | < 0.05 | NS | | > 0.05 |
| Ln $E_{400}$ | 0.140 | + | <0.001 | 0.142 | + | < 0.01 | 0.04 | + | > 0.05 | NS | | > 0.05 |
| Ln $vpdL_{400}$ | 0.281 | - | <0.0001 | NS | | > 0.05 | 0.119 | - | <0.0001 | NS | | > 0.05 |
| sqrt $g_{s400}$ | 0.414 | + | <0.0001 | 0.119 | + | < 0.01 | 0.142 | + | <0.01 | NS | | > 0.05 |
| Ln $WUE_i$ | 0.481 | - | <0.0001 | 0.420 | - | <0.0001 | 0.206 | - | <0.0001 | 0.357 | - | <0.0001 |
| $C_i/C_a$ | 0.332 | + | <0.0001 | 0.337 | + | <0.0001 | 0.192 | + | <0.0001 | 0.380 | + | <0.0001 |
| $C_{i400}$ | 0.352 | + | <0.0001 | 0.439 | + | <0.0001 | 0.199 | + | <0.0001 | 0.443 | + | <0.0001 |
| LDMC | 0.349 | - | <0.0001 | 0.331 | - | <0.0001 | 0.277 | - | <0.0001 | 0.366 | - | <0.0001 |
| Ln LMA | 0.569 | - | <0.0001 | 0.534 | - | <0.0001 | 0.412 | - | <0.0001 | 0.504 | - | <0.0001 |
| Ln FMA | 0.532 | - | <0.0001 | 0.516 | - | <0.0001 | 0.369 | - | <0.0001 | 0.452 | - | <0.0001 |
| Ln Leaf $N_{mass}$ | 0.192 | + | <0.001 | 0.264 | + | <0.001 | NS | | > 0.05 | 0.302 | + | <0.0001 |
| Ln Leaf $P_{mass}$ | 0.259 | + | <0.0001 | NS | | > 0.05 | NS | | > 0.05 | NS | | > 0.05 |
| Ln Ratio N/P | 0.102 | - | <0.01 | 0.167 | + | < 0.01 | NS | | > 0.05 | 0.09 | + | < 0.05 |
| Ln Leaf $N_{area}$ | 0.178 | - | <0.0001 | 0.269 | - | <0.001 | 0.363 | - | <0.0001 | 0.204 | - | <0.01 |
| Ln Leaf $P_{area}$ | N/S | | > 0.05 | 0.399 | - | <0.0001 | 0.228 | - | <0.001 | 0.286 | - | <0.0001 |
| $A_{400}$.N | 0.403 | + | <0.0001 | 0.162 | + | <0.01 | 0.396 | + | <0.0001 | NS | | > 0.05 |
| Ln $A_{400}$.P | 0.106 | + | < 0.01 | 0.272 | + | <0.001 | 0.334 | + | <0.001 | 0.119 | + | <0.01 |

17 NS = Non-significant
18
19
20
21

**Table S4: Variation in $\Delta^{13}$C and $WUE_i$ by biome.**

Mean and s.e of wet and dry season $\Delta^{13}$C and $WUE_i$. Means followed by different letters across sites in each season are significantly different (Tukey HSD, confidence level of 0.05).

| Site | $\Delta^{13}$C (‰) | | $WUE_i$ (‰) | |
|---|---|---|---|---|
| | Dry-season | Wet-season | Dry-season | Wet-season |
| SW | 19.75±0.49c | 19.15±0.32a | 78.14±5.26a | 84.60±4.30a |
| STS | 19.82±0.25c | 20.15±0.40a | 77.35±2.74a | 73.85±3.45a |
| TW | 20.16±0.25c | 20.18±0.28a | 73.84±4.10a | 73.57±3.03a |
| HTS | 20.77±0.38b,c | - | 67.21±2.84a,b | - |
| TWF | 21.54±0.55b,c | 23.51±0.56b | 68.80±5.91a,b | 37.68±6.12a,b |
| LTR | 22.98±0.46a,b | 23.02±0.39b | 43.25±4.93a,b | 42.86±4.17a,b |
| UTR | 24.05±0.76a | 24.31±0.52b | 31.74±8.16a | 29.04±5.62b |

31    **Calperum Mallee**

32    The Calperum Mallee SuperSite is in the mallee semi-arid ecosystem located approximately 25 km

33    north of Renmark in South Australia.  The landscape is an extensive plain with undulating mallee

34    woodland and riverine vegetation that fringes the River Murray and its anabranches. The vegetation

35    is dominated by upper storey Eucalypt trees of four species (*Eucalyptus dumosa, Eucalyptus*

36    *incrassata, Eucalyptus oleosa* and *Eucalyptus socialis*) (Meyer *et al.*, 2015).

37    **Great Western Woodlands**

38    The Great Western Woodlands located in south-west Western Australia is the largest remaining

39    intact semi-arid temperate woodland in the world. The vegetation comprises a 16-million hectare

40    mosaic of mallee, scrub–heath and woodland and is locally determined by edaphic factors and

41    influenced by historic disturbances (Gosper *et al.*, 2013). Mean annual rainfall is ~250 mm with the

42    highest-mean rainfall months in winter. *Eucalyptus salubris* constructs the dominant crown layer in

43    association with other *Eucalyptus* species (*E. salmonophloia, E. longicornis and E. moderata)* (Gosper

44    *et al.*, 2013).

45    **Alice Mulga**

46    The semi-arid Alice Mulga SuperSite is located approximately 200 km north of Alice Springs, in the

47    Northern Territory of Australia. The climate is characterized as having hot summers and warm

48    winters. Mean annual rainfall is ~300 mm and is highly seasonal, mostly occurring in large rainfall

49    events during summer. Vegetation is dominated by Mulga (*Acacia aneura* and related species)

50    woodlands, occasionally with large areas of spinifex under sparse woodland of *Corymbia* and other

51    *Acacia* species (Cleverly et al., 2016).

52    **Cumberland Plain**

53    The Cumberland Plain is a sclerophyll *Eucalyptus* woodland west of Richmond in New South Wales.

54    The soil is characterized by nutrient-poor alluvium from sandstone and shale bedrock in the Blue

55    Mountains deposited by the Nepean River. Despite being nutrient poor, this SuperSite supports high

56    regional biodiversity and endemic biota and is dominated by *Eucalyptus fibrosa*, *E. moluccana* and *E.*

57    *tereticornis* in the overstorey. Mean annual precipitation of this site is 900 mm (Table 1).

58 **Warra Tall Eucalypt**

59 The Warra Tall Eucalypt SuperSite is a cool, wet temperate forest located in Tasmania. Vegetation is

60 dominated by tall *Eucalyptus obliqua* occurring in a full range of successional stages from young

61 regrowth forests to old-growth mixed forests (Hickey *et al.*, 1999). Mean annual temperature at this

62 site is the lowest (~10°C), with a mean annual precipitation of 1474 mm (Table 1).

63 **Litchfield Savanna**

64 The Litchfield Savanna SuperSite is a ~1.5 km$^2$ tropical savanna 70 km south of Darwin in northern

65 Australia. This site is representative of the dominant ecosystem of that region. Climate of this site is

66 typical of northern Australia with extremely seasonal and high rainfall and approximately 56% of this

67 site is burnt annually (Murphy *et al.*, 2010). However, in this study, data collected from Howard

68 Springs (approximately 65 km north of Litchfield SuperSite; (Cernusak *et al.*, 2011) have been used as

69 a representative of this particular SuperSite. This approach is justified because both of these sites

70 had very similar vegetation and climate conditions as well as frequency of occurrence of fire. The

71 stand structure in these two sites are sufficiently similar as to not shift physiological properties at

72 the leaf-scale given the species occurred in both sites largely overlap (Bowman *et al.*, 2001; Hutley

73 and Beringer, 2010; Murphy *et al.*, 2010).

74 **FNQ Rainforest**

75 The FNQ Rainforest SuperSite is located in a tropical wet forest ~140 km north of Cairns in Far North

76 Queensland. This SuperSite is structurally divided into two transects – a) the lowland rainforest

77 based in the Daintree rainforest near Cape Tribulation (MAT = 25.2 °C, Ozflux site average MAP =

78 5700 mm) and b) the upland rainforest based around Robson Creek (MAT = 21 °C, MAP = 2140 mm).

79 Precipitation is highly seasonal with most occurring during summer (Weerasinghe *et al.*, 2014). FNQ

80 supports 10% of Australian flora despite of occurring in only 0.2% its landmass. Consequently a

81 substantial number of the species in this study comes from this SuperSite. Data from two nodes of

82 this SuperSite, i.e., Cape Tribulation and Robson Creek were collected and analysed independently in

83 this study because of significantly different environmental clines (altitude, MAT and MAP) that exists

84 in these two nodes of FNQ.

85

[Figure]

86

**Figure S1:**     **Location of SuperSites (represented by black dots).**

88

89

90

**Figure S2:**     **Mean annual temperature, mean annual precipitation, and biomes of the study sites.**

Each SuperSite is plotted in the Whittaker Biome Diagram (Whittaker, 1975) using the MAT and MAP observations generated for each site from the WorldClim data.

95

**(a)**                    **(b)**

[Figure]

96    **Figure S3:**    **Site mean values of (a) $\Delta^{13}C$ and (b) WUE$_i$**
97                      Dark and light bars represent mean of dry and wet-season respectively and the error bars
98                      represent standard errors.

[Figure]

**Figure S4: Ratio between intercellular and ambient [CO₂] (Cᵢ/Cₐ) for both seasons plotted as functions of mean annual precipitation (MAP).**

Left and right panels are plotted from dry- and wet-season data respectively. Statistically significant correlations with MAP are plotted with red regression lines.

[Figure]

[Figure]

**Figure S4:** Map of the Northern Territory (NT) of Australia and the study area within the NT and the Ti Tree basin.

---

## Author Response (AR2)

1) Please move Table S5 and add information

We have done this

2) Fig 2: Do we have stable isotope data from the stream/river?

Fig 2 does not incorporate/include data on stream/river water because the samples used in Fig 2 relate to GW isotopes only, and these are labelled as such (i.e. bore water). None of the trees sampled were close to water in a river/stream.

3) Fig 3 please show *E. camaldulensis* and *C. opaca* as different colours

We have done this

4) Figs 6, 7 please show *E. camaldulensis* and *C. opaca* as different colours

We have done this

5) p 31 line 1 – correct the omission of the phrase "differences in"

We have done this

6) Replace "Mulga" with A*c*acia spp.

We have done this

7) Map in the supplementary material is missing and numbers 1 – 4 definition.

Map is contained in the supplementary material and no numbers 1 – 4 included.

8) Does vertical distance mean elevation?

Yes it does and we have defined the word vertical to mean elevation above the stream.

9) Correct the conclusion regarding distance, increases and decreases in $\Delta^{13}C$ and WUEi.

We have corrected the error in the conclusion.

---

## Author Response (AR3)

**Response to final editorial comments**

1) Sampling plots and transect locations identified on the map

2) Deleted the reference to Supplementary Fig  S2 on page 6.

3) Clarified the meaning of the word "increase" in LVD with increasing DTGW  in line 31 of page 9.

4) Clarified the meaning of "positive effect of water supply on LVD" in line 19 of page 12.

5) Added missing column heading in Table 1.

6) Made required changes to Figures.

7) Corrected typo in figure legend of Fig. 6.

8) Defined abbreviations in Tables S3, S4 and site acronyms in Table S1.

9) Added acronyms to site descriptions in the Suppl. Material.

10) Defined AHD and green lines in map.